# Dual lysine and N-terminal acetyltransferases reveal the complexity underpinning protein acetylation

Willy V Bienvenut[1,†,‡] ID, Annika Brünje[2,†] ID, Jean-Baptiste Boyer[1] ID, Jens S Mühlenbeck[2] ID, Gautier Bernal[1,§] ID, Ines Lassowskat[2] ID, Cyril Dian[1] ID, Eric Linster[3] ID, Trinh V Dinh[3] ID, Minna M Koskela[4,¶] ID, Vincent Jung[1,††] ID, Julian Seidel[5] ID, Laura K Schyrba[2] ID, Aiste Ivanauskaite[4] ID, Jürgen Eirich[2] ID, Rüdiger Hell[3] ID, Dirk Schwarzer[5] ID, Paula Mulo[4] ID, Markus Wirtz[3] ID, Thierry Meinnel[1] ID, Carmela Giglione[1,*] ID & Iris Finkemeier[2,**] ID

## Abstract

Protein acetylation is a highly frequent protein modification. However, comparatively little is known about its enzymatic machinery. N-α-acetylation (NTA) and ε-lysine acetylation (KA) are known to be catalyzed by distinct families of enzymes (NATs and KATs, respectively), although the possibility that the same GCN5-related N-acetyltransferase (GNAT) can perform both functions has been debated. Here, we discovered a new family of plastid-localized GNATs, which possess a dual specificity. All characterized GNAT family members display a number of unique features. Quantitative mass spectrometry analyses revealed that these enzymes exhibit both distinct KA and relaxed NTA specificities. Furthermore, inactivation of GNAT2 leads to significant NTA or KA decreases of several plastid proteins, while proteins of other compartments were unaffected. The data indicate that these enzymes have specific protein targets and likely display partly redundant selectivity, increasing the robustness of the acetylation process *in vivo*. In summary, this study revealed a new layer of complexity in the machinery controlling this prevalent modification and suggests that other eukaryotic GNATs may also possess these previously underappreciated broader enzymatic activities.

**Keywords** acetylome; acetyltransferase; co- and post-translational modifications; plastid; quantitative proteomics

**Subject Categories** Plant Biology; Post-translational Modifications & Proteolysis; Proteomics
**Mol Syst Biol. (2020) 16: e9464**

## Introduction

Each single genome gives rise to myriads of dynamic proteomes. Protein modifications are mainly responsible for expanding the proteome inventory, playing countless functions important to guarantee full protein functionality (for reviews see Friso & van Wijk, 2015; Giglione *et al*, 2015; Aebersold & Mann, 2016). Among protein modifications, acetylation is one of the most common and intriguing. Two major types of protein acetylations have been identified thus far: N- α- and ε-lysine acetylation. Both modifications involve the transfer of an acetyl moiety from acetyl-coenzyme A (Ac-CoA), either to the α-amino group of the protein N-terminal amino acid or to the ε-amino group of lysines. However, N-terminal acetylation (NTA) and ε-lysine acetylation (KA) display a number of distinctive features. KA is a tightly regulated, reversible post-translational modification, whereas NTA is considered to be irreversible and to take place mainly co-translationally. In a few cases, this latter modification occurs post-translationally such as on actin by NAA80/NatH, on transmembrane proteins by NAA60/NatF, on hormone

1 Université Paris-Saclay, CEA, CNRS, Institute for Integrative Biology of the Cell (I2BC), Gif-sur-Yvette, France
2 Plant Physiology, Institute of Plant Biology and Biotechnology, University of Muenster, Muenster, Germany
3 Centre for Organismal Studies Heidelberg, University of Heidelberg, Heidelberg, Germany
4 Department of Biochemistry, Molecular Plant Biology, University of Turku, Turku, Finland
5 Interfaculty Institute of Biochemistry, University of Tübingen, Tübingen, Germany
*Corresponding author. Tel: +33 169829844; E-mail: carmela.giglione@i2bc.paris-saclay.fr
**Corresponding author. Tel: +49 251 8323805; E-mail: iris.finkemeier@uni-muenster.de
†These authors contributed equally to this work
‡Present address: Génétique Quantitative et Évolution, Gif-sur-Yvette, France
§Present address: Institute of Plant Sciences Paris-Saclay, Gif-sur-Yvette, France
¶Present address: Institute of Microbiology, Třeboň, Czech Republic
††Present address: Institute IMAGINE, Paris, France

peptides or in the maturation of exported proteins during plasmodium infection as well as in plastids of plants (Chang *et al*, 2008; Dinh *et al*, 2015 and references in Drazic *et al*, 2016; Aksnes *et al*, 2019). Both modifications are observed in all kingdoms of life. However, KA and NTA affect separately only 3–20% of all soluble proteins in prokaryotes and it was surmised that the frequency of these modifications increases with the complexity of the organism (Drazic *et al*, 2016). In multicellular organisms, KA occurs in the nucleus, cytosol, endoplasmic reticulum, mitochondria, and plastids much more frequently than NTA, which is mostly associated with cytoplasmic proteins (Varland *et al*, 2015; Linster & Wirtz, 2018). Nonetheless, several reports showed that NTA, together with KA (Hartl *et al*, 2017), is a widespread modification in chloroplasts, which occurs co-translationally on plastid-encoded proteins as well as post-translationally on a significant fraction of imported nuclear-encoded proteins after the cleavage of their transit peptides (Zybailov *et al*, 2008; Bienvenut *et al*, 2011, 2012; Bischof *et al*, 2011; Huesgen *et al*, 2013). Although the number of experimentally characterized N- and/or K-acetylated (NTAed and KAed) proteins is continuously increasing, many features of the acetyltransferases that catalyze KA and NTA are much less understood, particularly those that originate from prokaryotes and specifically operate in organelles such as mitochondria and chloroplasts.

All known N-terminal-α-acetyltransferases (NATs) belong to the superfamily of general control non-repressible 5 (GCN5)-related *N*-acetyltransferases (GNAT), whereas the identified lysine acetyltransferases (KATs) are grouped in at least three families: GNAT, MYST, and p300/CBP (Friedmann & Marmorstein, 2013; Montgomery *et al*, 2015; Drazic *et al*, 2016).

GNAT proteins are characterized by a low overall sequence homology (3–23%) but they display conserved secondary and 3D structures (Vetting *et al*, 2005). Although the GNAT domain has largely diverged, a general profile has been developed and used to identify proteins belonging to the GNAT superfamily, including NATs and KATs (Vetting *et al*, 2005; Hulo *et al*, 2008; Rathore *et al*, 2016; Salah Ud-Din *et al*, 2016).

In eukaryotes, several cytosolic NAT and KAT complexes are known with distinct substrate specificities, which are conserved throughout eukaryotic evolution (Drazic *et al*, 2016). The NAT specificity is generally defined by the first two amino acids of the substrates, despite the observed negative influence of distant residues (i.e., the K and P inhibitory effects within positions P′2–P′10) (Arnesen *et al*, 2009b; Hole *et al*, 2011; Van Damme *et al*, 2011). This is different to the identified prokaryotic NATs, which are composed of only a catalytic subunit, and which display restricted or extended substrate specificities in eubacteria and archaea, respectively (Giglione *et al*, 2015). In contrast to eukaryotic proteins and recent results (Christensen *et al*, 2018; Reverdy *et al*, 2018; Carabetta *et al*, 2019), it was believed that prokaryotic KA played only a minor role and not much attention has been given to the corresponding KAT machinery (for review, see (Christensen *et al*, 2019a,b)).

The consensual knowledge favors distinct enzymes for the acetylation of protein N-termini and lysine residues (Liszczak *et al*, 2011; Magin *et al*, 2016). The cytosolic acetyltransferases NAA40, NAA50, and NAA60 have been shown to display weak KA and strong NTA activities (Evjenth *et al*, 2009; Liu *et al*, 2009; Chu *et al*, 2011; Yang *et al*, 2011; Stove *et al*, 2016; Armbruster *et al*, 2020; Linster *et al*, 2020). However, there still is controversy on whether or not NAA10

might catalyze both reactions (Friedmann & Marmorstein, 2013; Magin *et al*, 2016). Indeed, several reports suggest that the catalytic subunit of the human and yeast NatA complex (Ard1/NAA10) is able to have both NTA and KA activities on specific substrates (Jeong *et al*, 2002; Arnesen *et al*, 2009a,b; Evjenth *et al*, 2009; Shin *et al*, 2009; Yoon *et al*, 2014). However, KA failed to be further confirmed for some of these substrates (Arnesen *et al*, 2005; Murray-Rust *et al*, 2006) and *in vitro* KA of other substrates was shown to be enzyme-independent and simply promoted by increasing concentrations of Ac-CoA (Magin *et al*, 2016; Aksnes *et al*, 2019). These studies argued against a role for NAA10 in KA and leave open the question of a double KAT/NAT activity for the same acetyltransferase. Interestingly, a new acetyltransferase has been described in the chloroplast of the model plant *Arabidopsis thaliana*, and, surprisingly, this enzyme displayed auto-KAT activity in addition to unusual promiscuous NAT activity (Dinh *et al*, 2015). Besides this first report and the recent identification of the plastid lysine acetyltransferase NSI in the chloroplast (Koskela *et al*, 2018), the plastid NAT and KAT machineries remain uncharacterized thus far.

In the current study, we sought to identify putative *Arabidopsis* NAT and/or KAT candidates using the PROSITE GNAT-associated profiles and the plastid subcellular localization prediction. Such investigation revealed 10 putative *Arabidopsis* GNAT proteins. Subcellular localization analyses in *Arabidopsis* protoplasts confirmed a plastid or plastid-associated localization for only eight of the 10 putative GNATs. Furthermore, by using the recently developed global acetylome profiling approach (Dinh *et al*, 2015), as well as a quantitative mass spectrometry-based lysine acetylome analysis (Lassowskat *et al*, 2017), we discovered that six of the eight GNATs display significant dual NAT and KAT activities. The remaining two candidates showed only weak KAT as well as NAT activities on a few substrates. All of the GNATs displaying an NTA activity exhibited extended NAT substrate specificities compared to the cytosolic ones. Proof of concept of the dual activity borne by one member was demonstrated in one of the GNAT knockout mutants where deficit of either acetylation levels was observed on plastid proteins. Altogether, this work identifies a new and widespread dual function for the acetyltransferases, which overturns conventional knowledge in this area, and therefore may have far-reaching implications for the study of acetylation in eukaryotic organisms.

## Results

### *In silico* analyses of the *Arabidopsis* genome revealed 10 GNAT enzymes with putative plastid localization

To identify new acetyltransferases responsible for protein acetylation in plastids, we searched the *Arabidopsis* genome for proteins, which possess both a GCN5-related *N*-acetyltransferases (GNAT) domain and a predicted organellar N-terminal transit peptide. Our final database search for putative NATs and KATs converged to 10 candidate proteins (Table EV1 and Dataset EV1). Two of these proteins have recently been identified in plastids of *Arabidopsis* as NAT (NAA70) and as KAT (NSI) enzymes, respectively (Dinh *et al*, 2015; Koskela *et al*, 2018).

Because the catalytic activity of these proteins (i.e., whether they transfer acetyl groups to protein N-termini, to internal lysine

residues of proteins, or to metabolites) cannot be predicted only from their amino acid sequence, we called these enzymes GNAT1–10 (Fig 1, Table EV1). To get some more insights into the

relationship between these diverse types of acetyltransferases, we constructed a phylogenetic distance tree including known GNAT proteins from *Arabidopsis*, yeast, and *Escherichia coli* (Figs 1A and

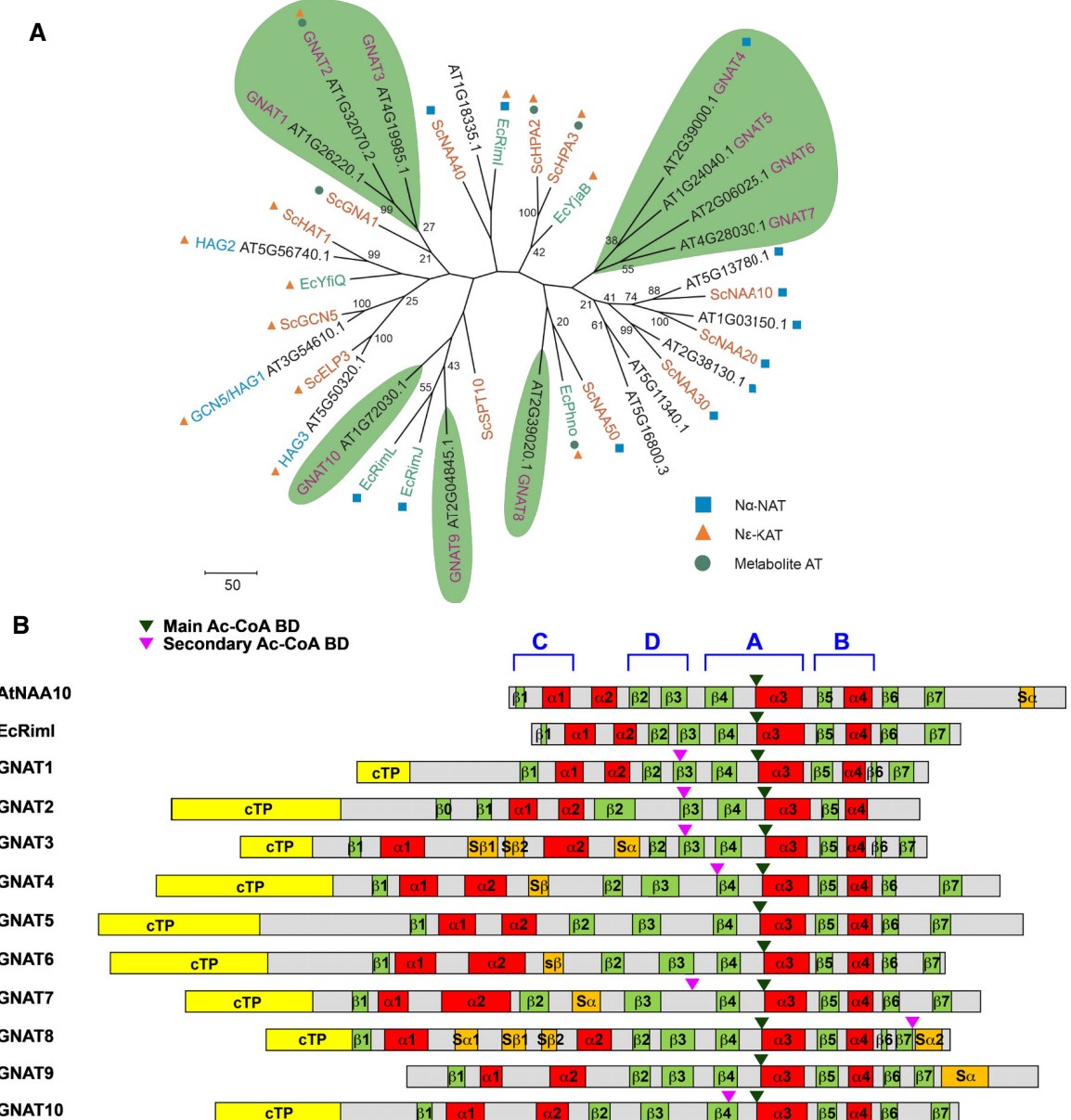

**Figure 1. Putative organellar KAT and NAT genes from *Arabidopsis*.**

A Phylogenetic tree of GNAT candidates from *Arabidopsis thaliana* (black letters), *Saccharomyces cerevisiae* (orange letters), and *Escherichia coli* (green letters) containing the acetyltransferase Pfam domains (PF0058, PF13302, PF13508, PF13673) (Finn *et al*, 2006). GNAT family sequences were aligned with ClustalW, and a phylogenetic tree was designed by applying the neighbor-joining method. Bootstrap analysis was performed using 2,000 replicates, whereby the resulting bootstrap values (values ≥ 20) are indicated next to the corresponding branches. The tree-specific topology was tested by maximum parsimony analysis. GNAT candidates with a putative organellar localization (TargetP1.1) were highlighted with a green background and named as GNAT1 to GNAT10 in relation to their position in the phylogenetic tree. Squares, triangles, and circles describe the specific acetylation activity, which was reported in literature. The metabolic activity of GNAT2 corresponds to serotonin acetyltransferase (Lee *et al*, 2014).

B Schematic overview of organellar GNATs' secondary structure organization (including AtNAA and EcRiml for comparison). Secondary structural elements of the GNAT candidates were determined using Jpred tools in combination with structure homology models (Swiss-model) and are displayed in red (α-helixes), green (β-strands), and orange (supplementary secondary elements). All candidates were predicted with a mitochondrial or a chloroplastic transit peptide (cTP) using TargetP. The mature form of these candidates is released after the excision of this cTP. Positions of main and secondary Acyl-CoA binding domain (Ac-CoA BD) are shown. C, D, A, and B design the four conserved motifs comprising what is referred as the *N*-acetyltransferase domain (Dyda *et al*, 2000).

EV1). *Arabidopsis* GNAT1–3 cluster together with known histone-acetyltransferase (HAT) proteins from *Arabidopsis* and yeast (Fig 1A) and defined a first subtype of GNAT-related sequences (subtype 1, Fig EV1). GNAT4, 5, 6, 7, and 10 are located on a distinct branch (subtype 2, Fig EV1) and finally GNAT8 and GNAT9 group into a third subtype (Fig EV1).

Proteins of the GNAT superfamily have an overall low primary sequence similarity. However, all GNAT members display a conserved core of six to seven β-sheets and four α-helixes ordered as β0–β1–α1–α2–β2–β3–β4–α3–β5–α4–β6 (Salah Ud-Din *et al*, 2016). These secondary structural elements arrange in four conserved motifs (A–D; Fig 1B). The A and B domains are involved in Ac-CoA interaction and acceptor substrate binding, respectively. The C and D domains are involved in protein stability (see for review Salah Ud-Din *et al*, 2016). Although this pattern fits to most of the members of the GNAT superfamily, some deviations were identified such as the missing α2-helix in the HATs, or additional elements, e.g., the additional α-helix between β1 and β2, in the *Enterococcus faecalis* aminoglycoside 6′-*N*-acetyltransferase (Wybenga-Groot *et al*, 1999).

The JPred tools (Drozdetskiy *et al*, 2015) together with homology models obtained from Swiss-model were used to predict the secondary structural elements of the 10 selected GNATs (Fig 1B). All candidates exhibit the typical GNAT topology with several clear primary sequence divergences from the cytosolic catalytic NAT enzymes (Dataset EV1). Particularly, both β1 and β2 strands are poorly conserved in their sequences. Moreover, the length, number, and position of the α-helices between the N-terminal β1 and β2 strands reveal some dissimilarities among the different GNATs (Fig 1B and Dataset EV1). Because variation in this region can reflect differences in substrate specificity (i.e., allowing the formation of different acceptor substrate binding sites), we anticipated a different substrate specificity for our candidates. In addition, a poor sequence conservation was also retrieved among the different GNATs at the level of the secondary elements forming the binding site for the acceptor substrate (loop between β1- and β2-strands, α4-helix and β7-strand), similar to other members of the GNAT superfamily (Salah Ud-Din *et al*, 2016). Still a number of residues were found remarkably conserved across all selected GNATs (Dataset EV1).

The GNAT Ac-CoA binding domain (BD) is generally located at the N-terminal side of the α3-helix. This specific and crucial domain for the acetyltransferase activity shares sequence homology over species with some similarity to the ATP-BD P-loop. For GNATs, the proposed conserved "P-loop like" sequence is [QR]-x-x-G-x-[GA] (Salah Ud-Din *et al*, 2016), where x could be any amino acid. An enlarged version of this pattern, including L at position 9, was proposed ([QR]-x-x-G-x-[GA]-x-x-L) for eukaryotic NATs (Rathore *et al*, 2016) and also observed in *Staphylococcus aureus* GNAT superfamily members (Srivastava *et al*, 2014). Surprisingly, Cort and co-workers (Cort *et al*, 2008) reported a major variation for one of the *S. aureus* GNAT superfamily (SA$_{COL}$2532) with a G instead of the expected Q/R residue at position 1. A similar variability was observed by Rathore *et al* (2016), suggesting possible divergences at position 4. Investigation of the consensus "P-loop like" in the putative GNATs clearly showed unique features with a slight degeneration of the conserved sequence for few of them (Table EV2). To verify whether the divergences observed in the Ac-CoA BD were only species-specific, we performed a larger scale orthologue

investigation. This approach confirmed the previously mentioned divergences and highlighted some new conserved sites (Table EV2). It appears that the residue at position 5 and 10 retains some specificity associated with hydrophobic residues including L/I/M/V. From this investigation, we could establish an Ac-CoA BD consensus pattern for each of the putative GNATs and a new enlarged version of this pattern corresponding to [RQ]xxG[LIMV][AG]xx[LIMVF][LIMV] (Table EV2).

We also observed that seven of the GNAT candidates possess more than one Ac-CoA BD (Table EV2 and Fig 1B). These duplicated "P-loop like" sequences display a degenerated pattern on the residues at positions 5, 9, and 10 (Table EV2) and are extremely rare in cytoplasmic NATs. Out of these multiple Ac-CoA BD, the most conserved ones (labeled as main Ac-CoA BD) were usually located at the N-terminus of the α3-helix as reported for other GNATs (Fig 1B).

Several residues previously shown to be involved in substrate binding and specificity in cytosolic NATs (Liszczak *et al*, 2011, 2013) were also found to be conserved in some of the GNATs (Dataset EV1). For instance, in HsNAA50 the two catalytic residues Y73 in β4 strand and H112 residue in β5 strand, which are representative of the general base positions in GNAT enzymes (Liszczak *et al*, 2013), are found in GNAT4, 5, 6, and 7 and, to a lesser extent in GNAT9 and 10, in which the equivalent H112 is replaced by E and Y residues, respectively. Similarly, GNAT8 displays a catalytic dyad equivalent to HsNAA30 at these same positions (Y283 in β4 strand and E321 residue in β5 strand). Interestingly, the β4/β5 catalytic dyad is not conserved in GNAT1, 2, and 3. Indeed, Y73 was replaced by I in GNAT1, T in GNAT2, and S in GNAT3, whereas H112 is replaced by F in GNAT2 and by Y in GNAT1 and 3 suggesting that the catalytic dyads in these GNATs are either different or positioned differently on the strands. Finally, several other highly conserved residues especially in α4 helix, such as the Y124 residue of HsNAA50, are found in all *N*-α acetyltransferases except for GNAT3 and 9, where the Y is replaced by F (Dataset EV1).

## Seven of the 10 predicted *Arabidopsis* GNATs are localized within plastids

To confirm the predicted plastid localization (Table EV1), all GNAT candidate proteins were expressed in *Arabidopsis* protoplasts as fusion proteins with a C-terminal GFP-tag under a 35S-promoter (Fig EV2). An overlapping GFP and chlorophyll autofluorescence confirmed plastid localizations of GNAT1, 2, 3, 4, 5, 7, and 10. The GNAT6-GFP showed a spotted fluorescence pattern, which was found either associated with chloroplasts or confined within the nuclear envelope (Figs EV2 and EV3A–C). Mitotracker staining revealed no overlap of the GNAT6-GFP fluorescence with mitochondria (Fig EV3D). The fluorescence signal of GNAT8- and 9-GFP expressing protoplasts was similar to those of the free GFP, which indicates cytosolic/nuclear localization. GENEVESTIGATOR publicly available gene expression data highlighted that all plastid-localized GNATs are mainly expressed in green tissues, GNAT6 is also expressed in roots, whereas GNAT8 and 9 cluster in a separate gene expression group and are expressed throughout the plant (Fig EV4). As GNAT8 and 9 showed a clear cytosolic and non-plastid-related localization, and considering their clustering to a different subtype (Fig EV1), we excluded them from further investigations.

## Recombinant GNAT2 and GNAT10 display both KA and NTA activities *in vitro*

Sequence and structural analyses of the plastid or plastid-associated GNATs and previous studies (Dinh *et al*, 2015; Koskela *et al*, 2018) suggested that some of those GNATs might behave as NATs or/and KATs, despite a number of peculiarities observed in these members. Therefore, to assess whether they could display acetyltransferase activities, we selected one member of each of the two subtypes (Fig EV1). GNAT2 as archetype of subtype 1 and GNAT10 as the archetype of subtype 2 could be obtained as soluble and stable fusion proteins with a N-terminal His-tag, and including a maltose-binding protein, from *E. coli* extracts. We used a HPLC-based enzyme assay taking advantage of a series of designed peptides as substrates. These peptides are derived from an established acetylation enzyme assay (Seidel *et al*, 2016; Koskela *et al*, 2018) and display either a free amino group (A, G, S, T, V, M, or L) at the N-terminus followed by the sequence AQGAK(ac)AA-R or alternatively an acetylated N-terminus and a free ε-amino group on the internal lysine side chain. These modifications allowed to unambiguously assay either NTA or KA activity. To compare the activities of both enzymes with the different peptides, we used fixed substrate and protein concentrations in the enzyme assay (Table 1). Both purified enzymes unambiguously catalyzed dual KA and NTA activities, demonstrating that both GNATs were sufficient to carry out the two acetylation activities and that they do not require a further regulatory subunit. In addition, our peptide series with varying amino acid at position 1, even if not exhaustive, suggested that both GNATs have relaxed substrate specificities for NTA. However, GNAT2 showed a NTA preference for V and T as substrate, while GNAT10 showed a one order of magnitude higher NTA activity on the alpha-amino group of L, M, and V compared to A. These conclusions with GNAT10 are fully consistent with those we previously obtained with GNAT4, another member of GNAT subtype 2 (see Supplemental figure 3 in Dinh *et al*, 2015). Additionally, these data indicate that

the chemical properties of the N-terminal amino acid have an impact on the NTA activity of the two GNATs.

To unravel the preferred substrates for KA and NTA of the entire plastid GNAT family in an unbiased manner, we designed an assay using the *E. coli* proteome as random putative substrates when one of the eight selected GNATs was expressed. The results are detailed in the two paragraphs below.

## Plastid GNATs possess large-spectrum KA activity

To fully characterize the KA activities of all plastid and plastid-associated GNATs, we undertook a global K-acetylome analysis. First, we expressed all eight GNAT proteins without their predicted target sequences, but with an N-terminal His-tag and a maltose-binding protein, under a T7 promoter in *E. coli*. All proteins were clearly overexpressed after induction of transcription with IPTG (Fig 2A). To assess whether the recombinant proteins can act as KAT enzymes, we first used an anti-acetyllysine antibody to detect acetylated proteins in the *E. coli* extracts (Fig 2A). Under non-induced conditions, a clear pattern of KA on *E. coli* proteins was observed in the Western blot analysis, similar to those from other reports (Christensen *et al*, 2018). Upon induction of the GNAT expression, we observed a K-hyperacetylation in almost all GNAT recombinant proteins (Fig 2A). A clear and strong hyperacetylation of *E. coli* proteins was detected when GNAT4 was overexpressed. This is interesting as a NTA activity was previously reported for GNAT4, in addition to some autoacetylation activity on internal lysine residues (Dinh *et al*, 2015). Since the Western blot analysis with the anti-acetyllysine antibody is not very sensitive and does not reveal the identity of the acetylated proteins and sites, we determined the KA proteins before and after induction of the respective GNATs with quantitative mass spectrometry (Dataset EV2). From around 3,000 quantified KA sites in the *E. coli* extracts from two biological replicates for each GNAT transgenic strain (Dataset EV2A), eight to 201 new KA sites were detected only upon overexpression of the respective *Arabidopsis* GNATs and not

**Table 1.  GNAT2 and GNAT10 exhibit dual KA and NTA activities *in vitro*.**

| Acetylation type | Residue or group at positions 1 or 6 in peptide[a] $X_1AQGAK_6AA$[b] | | Enzyme-catalyzed *N*-acetylation position assayed | NAT activity (nmol[NTAed-peptide]·min$^{-1}$·µmol [GNAT]$^{-1}$)[c] | |
|---|---|---|---|---|---|
| | 1 | 6 | | GNAT2 | GNAT10 |
| NTA | A | Ac-K | 1 | 11.6 ± 0.4 | 17.2 ± 0.1 |
| | G | Ac-K | 1 | 18.4 ± 0.5 | 38.4 ± 0.4 |
| | S | Ac-K | 1 | 21.2 ± 0.4 | 40.1 ± 0.2 |
| | T | Ac-K | 1 | 31.8 ± 1.3 | 92.5 ± 0.3 |
| | V | Ac-K | 1 | 34.4 ± 5.3 | 185.8 ± 0.3 |
| | M | Ac-K | 1 | 11.8 ± 2.3 | 192.4 ± 0.5 |
| | L | Ac-K | 1 | 7.9 ± 5.6 | 324.3 ± 0.2 |
| KA | Ac | K | 6 | 7.0 ± 0.1 | 3.0 ± 0.1 |

[a]Peptide concentration was 50 µM.
[b]Full peptide sequence is given in Materials and Methods. Ac is for *N*-acetyl.
[c]Enzyme catalysis was started by the addition of 100 µM acetyl-CoA to the reaction mixture already containing the indicated GNAT (5 µM). Incubation was for 45 min at 30°C. NTA was assessed by reverse-phase HPLC (*n* = 3, ± SD).

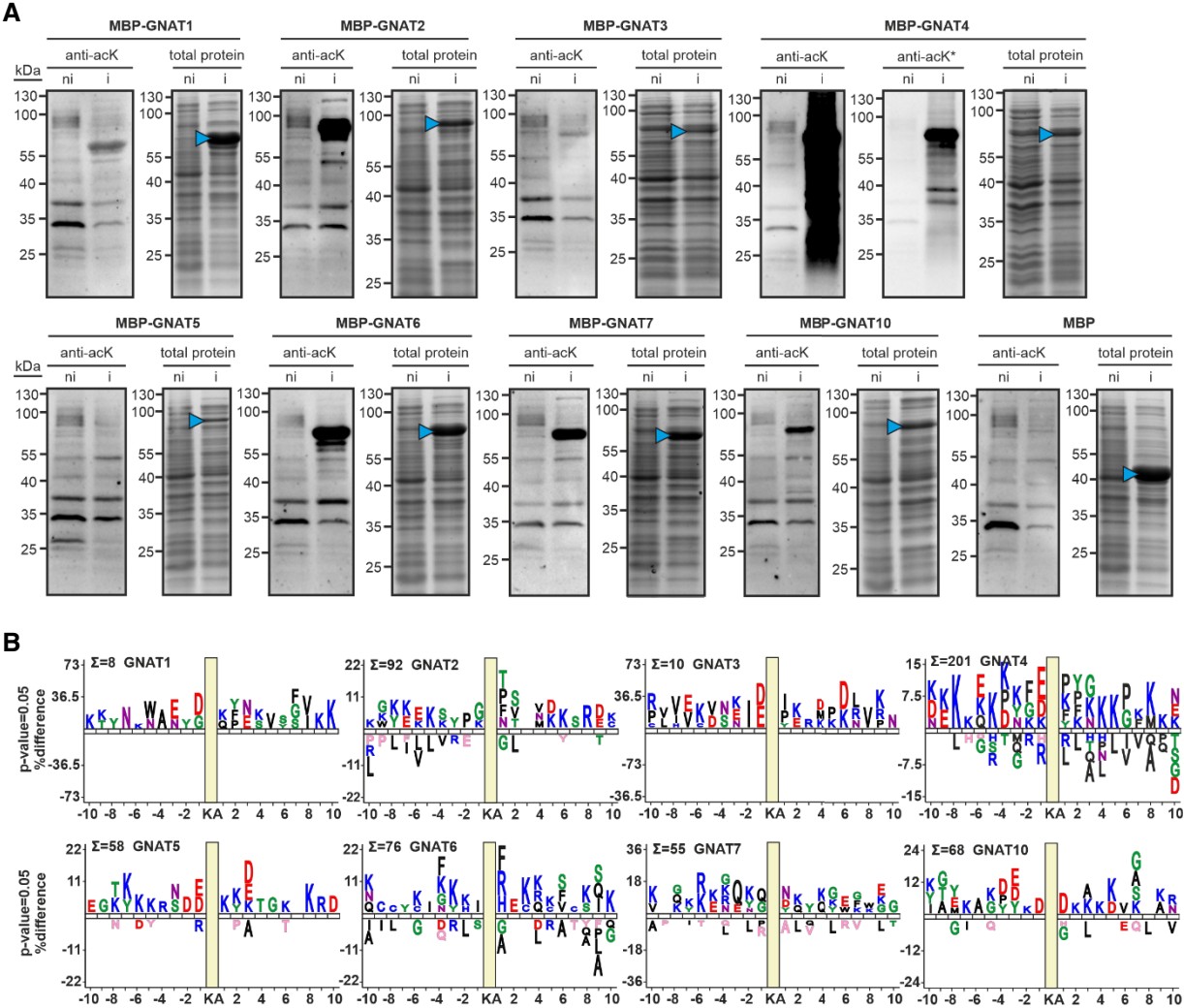

**Figure 2. Lysine acetylation is induced in *Escherichia coli* extracts upon *Arabidopsis* GNAT expression.**

A    Anti-acetyllysine Western blot analyses and total protein stains of *E. coli* cell extracts. Cellular protein extracts before and after induction of His6-MBP-GNAT expression were separated on 12% acrylamide gels and immunoblotted by using an anti-acetyllysine antibody or stained with Coomassie dye. GNAT1, 2, 5, 6, 7, and 10 as well as His6-MBP control were expressed in *E. coli* BL21(DE3)pLysS. GNAT3 and 4 were expressed in Rosetta(DE3). In the total protein stains, recombinant GNAT protein constructs were highlighted by blue arrowheads. Protein expression before (ni) and after (i) induction with IPTG is indicated. The luminescence signal, indicating acetylated proteins, was usually recorded after 40–120 s. Since MBP-GNAT4 expression resulted in a saturated signal, the luminescence was additionally recorded for 10 s (indicated by an asterisk).

B    Sequence logos of all unique lysine acetylation sites after GNAT expression in *E. coli*. Uniprot *E. coli* protein sequences were used as background population (sequence logos were generated using iceLogo (Maddelein *et al*, 2015)).

in the empty vector expressing strains as control (Fig 2B, Dataset EV2B–Q). These newly acetylated proteins were either proteins from *E. coli* or the recombinant GNAT proteins themselves. Especially the MBP-tag was highly acetylated on most of the fusion proteins, which might be due to the unusually high abundance of MBP during overexpression. In general, the GNAT induced KA of the endogenous *E. coli* proteins did not affect the overall abundance of most of these proteins, except for a few cases (Fig EV5). For example, the *E. coli* transketolase (P33570) was acetylated by GNAT5, 7, and 10 at K396, which coincided with a more than two-fold increase in protein abundance in all three strains (Dataset EV2). From the global K-acetylome profiling, we can conclude that

all GNATs have a KA activity on *E. coli* proteins, although GNAT1 and 3 showed the least number of protein substrates. In agreement with the Western blot analysis, GNAT4 overexpression resulted in the highest number of new KA sites on the *E. coli* proteome in comparison with all other tested GNATs (Figs 2 and EV5 and Dataset EV2). Interestingly, the GNATs showed very distinct substrate specificities, since there was not much overlap in the identity of the target proteins between the GNATs, nor in the 10 amino acids surrounding the acetylated lysine residues (Dataset EV2 and Fig 2B). Similar to the previously reported sequence logo from the lysine-acetylated peptides identified on chloroplast proteins of *Arabidopsis* (Hartl *et al*, 2017), only GNAT 1, 3, 4, 5,

and 10 preferred to some extend acidic amino acids in the -1 position next to the KA residue.

## Global N-α-acetylome profiling unravels the relaxed NTA substrate specificity of plastid GNATs

To complete the characterization of this new GNAT family, we assessed their potential NTA activity and substrate specificity. This investigation was performed using the Global Acetylation Profiling (GAP) assay (Dinh *et al*, 2015) based on the recombinant expression of the GNATs in *E. coli*. Similar to the K-acetylome profiling analysis, this *in cellulo* approach provides the NTA characterization of the *E. coli* protein N-termini exclusively dependent on the putative expressed protein (Bienvenut *et al*, 2017a,b). This approach has been previously validated, confirming for instance the substrate specificity of the cytosolic *At*NAA10 (Dinh *et al*, 2015; Linster *et al*, 2015) for N-terminal residues such as A, T, S, G, or V, unmasked after the removal of the first M by MetAP enzymes (Giglione *et al*, 2015). The GAP assay is very sensitive especially as the natural prevalence of NAT is very low in *E. coli*, with only few significantly acetylated proteins. These endogenously acetylated proteins are duly cataloged (Bienvenut *et al*, 2015; Schmidt *et al*, 2016) and were excluded from the analysis below (see Dataset EV3).

The complete list of 397 protein N-termini revealed by the GAP assay for the eight GNATs is compiled in Dataset EV3. As shown in Fig 3A, six out of the eight GNATs were able to significantly and specifically increase the number of NTAed substrates in the *E. coli* proteome. Only very few substrates were retrieved mostly with GNAT1 but also with GNAT3 (Fig 3A). In addition, because the few substrates of GNAT3 were only moderately *N*-acetylated (mainly below 20%, Fig 3A), GNAT1 and GNAT3 do not appear to be efficient for protein NTA and KA. Both GNATs most likely act only either (i) on a restricted number of specific plastid substrates, which are absent from the *E. coli* proteome, or (ii) they require plant-specific accessory proteins, like cytosolic NATs, to improve their activity. In contrast, GNAT4, 6, and 7 provided the largest number of modified N-termini (Fig 3A). Furthermore, an important fraction of the characterized substrates (55, 58, and 54%, respectively) could be quantified with an increase of the NTA yield higher than 20% (Fig 3A). GNAT2, 5, and 10 defined a second rank for the number of characterized substrates, with GNAT10 providing the lowest numbers and the lower NTA yield increase

(Fig 3A). Note that the relative expression level of each GNAT does not influence the NTA and KA activity as indicated in Fig 3A. For instance, GNAT1 is the best expressed GNAT in the assay but it modifies the lowest number of proteins with high efficiency (see dashed line in Fig 3A); this is in contrast to GNAT5.

Unlike cytosolic NATs, all GNAT candidates displayed a relaxed NTA substrate specificity but with significant differences in terms of favored substrates (Figs 3B and 4 and Dataset EV3). For instance, all of the six most active plastid-associated GNATs were very efficient for N-termini starting with an initiator methionine (iMet), but with clear differences induced by the amino acid at position 2 (Figs 3B and EV6). Additionally, GNAT2, 4, 5, and 7 were more efficient to act on NatA-like substrates (i.e., A, S, T, etc.) in comparison with GNAT10, which in turn seemed to prefer NatC/E/F- and to a lesser extent NatB-like substrates (M starting proteins; Figs 3B and EV6). Finally, GNAT6 was found to be almost equally active on NatA-, NatB-, and NatC/E/-like substrates (Figs 3B and EV6). Besides, the analysis of all GAP data (Dataset EV3) revealed that despite a few sequence motifs that were NTAed by all GNATs, some other sequences were only recognized by one GNAT and not by the others, while some others were recognized by several GNATs (Fig 4A and Dataset EV3). Finally, building a logo representation of specific GNAT2 substrates, over that of the other seven GNATs, revealed that its substrate specificity is not only based on the first amino acid but also on the following residues with preference for neutral and small amino acids (Fig 4B, and next paragraph).

## GNAT2 knockout lines identify NTAed and KAed plastid targets demonstrating dual acetylation activity *in vivo*

As a proof of concept that a GNAT of this family does display both NTA and KA activities *in planta*, we anticipated that gene disruption of one of the members might lead to lower acetylation yields of both types of targets. Interestingly, in a previous study, we have shown that GNAT2 (NSI) knockout in two independent backgrounds (*nsi-1* and *nsi-2*) led to a clear phenotype with defective state transitions in the chloroplast (Koskela *et al*, 2018). Lysine acetylome analysis revealed a decreased KA status of several components of the photosynthetic apparatus, while the overall protein abundance was unchanged (Koskela *et al*, 2018). From these data, it appeared that

---

**Figure 3. Global acetylome profiling reveals relaxed NTA specificity of six out of the eight GNATs.**

Qualitative and quantitative MS-based analysis was performed with the GAP assay as described previously (Dinh *et al*, 2015). NTA yields were quantified as previously reported (Bienvenut *et al*, 2017b). Expression of each GNAT was ensured from a plasmid vector. A subset of thirty-two proteins naturally NTAed (i.e., > 20%) in the various samples and expressing the empty vector was disregarded (see Bienvenut *et al*, 2015 for details of such a dataset).

A   Number of characterized substrates with significant increases of NTA yields. The NTA yields of all retrieved N-termini upon a given GNAT expression were divided into four categories and each column is the result of the sum of all categories: 50–100%, dark green; 20–50%, light green; 10–20%, orange; 5–10%, red. The number of protein occurrence with a yield higher than 30% is also displayed to visualize the most significant acetylation status of each GNAT. The respective overexpression level of each GNAT as assessed from gel electrophoresis displayed in Fig 2A is indicated below (++++ is for very high expression, +++ for high, ++ for medium, + for intermediate).

B   Overview of GNAT substrate specificities. The number of NTAed substrates with each of the six major active GNATs is displayed in a pie chart according to the N-terminal pattern retrieved. The dataset is identical to that of panel A. N-terminal residues of proteins starting with Met are displayed in dark and light orange while all the other likely resulting from NME are colored with a blue pallet. Very similar pictures are obtained if only N-termini with NTA yields over 50% are figured. The starting, acetylated amino acid is indicated and labeled with the cytosolic Nat-type substrate specificity (i.e., NatA-F) according to the classic distribution available in Aksnes *et al* (2016). Details of the fraction of N-termini starting with M are given in Fig EV6.

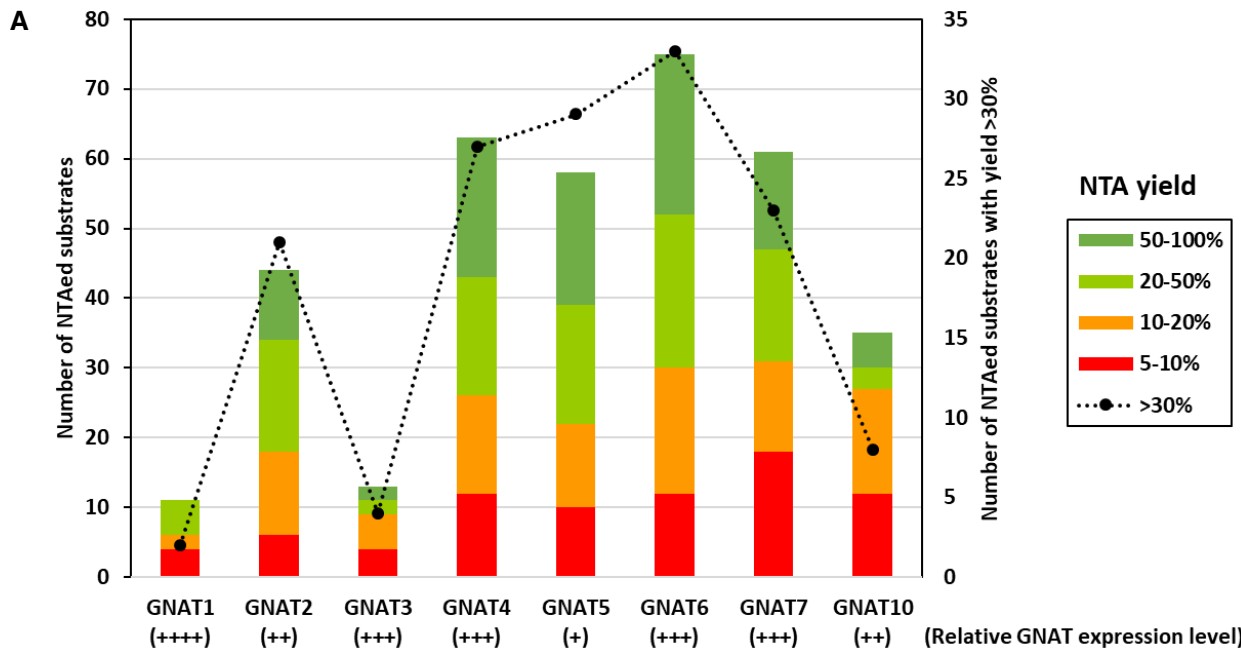

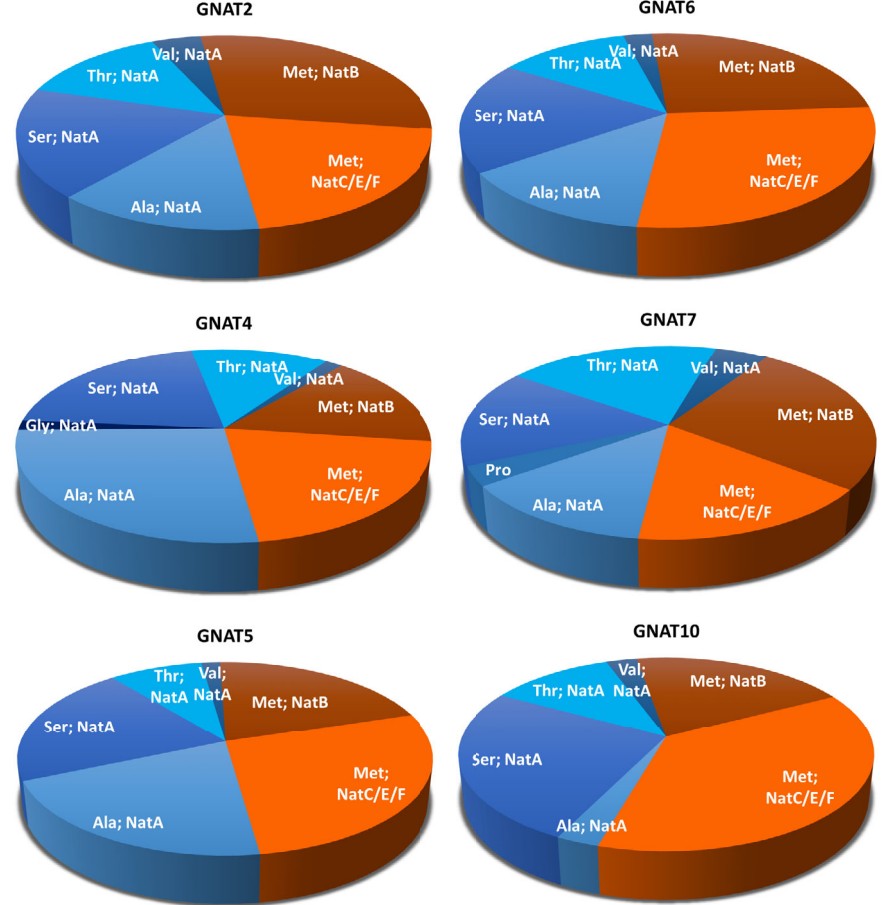

**Figure 3.**

at least GNAT2 displayed a unique KA substrate specificity, which could not be compensated by the other GNAT members. Because of the discovered dual KA and NTA activity, we therefore investigated whether NSI-defective plant lines were also affected in their overall NTA status using tissue from the same growth conditions as analyzed before (Koskela *et al*, 2018). Hence, we performed a global NTA quantitative analysis on WT and mutant lines using the SILProNAQ procedure as described previously (Linster *et al*, 2015; Frottin *et al*, 2016; Huber *et al*, 2020). We characterized almost 2,000 N-termini in various samples and could quantify half of them in four replicates of each line (Fig 5A and B and Dataset EV4).

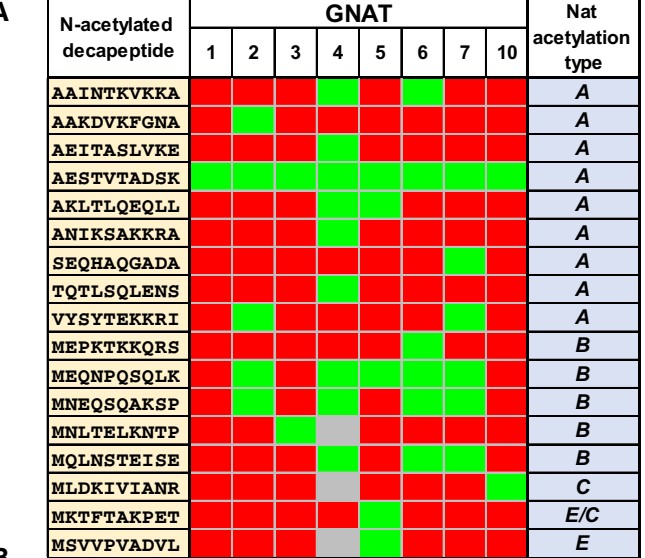

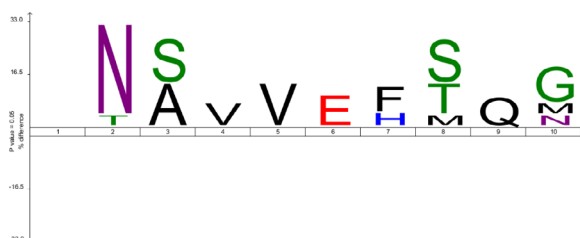

**Figure 4. GNAT2, like all plastid GNATs, has redundant and specific selectivity profiles.**

A  Table showing examples of the selectivity of all plastid GNATs on a selection of retrieved N-termini. The table is extracted from Dataset EV3 to illustrate the concept and is not exhaustive. The 10 first amino acids of the N-α-acetylated proteins are indicated. The color code is green, positive; red, negative; gray means that the data are missing, i.e., that the peptide was not quantified.

B  IceLogo representation (Colaert *et al*, 2009) of the N-termini substrates of GNAT2 vs all that of other GNATs. To construct the dataset, all GNAT2 substrates with NTA yield threshold > 30% were selected in the positive set. The negative set corresponded to the compilation of all substrates with a threshold > 30% of all other seven GNATs. The color symbol is associated with the default choice, which the software proposes for each class of amino acid: green is for the class of small hydrophilic uncharged residues including S, T, G. Acidic residues including D or E are colored in red. Positively charged residues including K, R, and H are displayed in blue. N or Q is in purple. Hydrophobic residues (A, I, M, V, L, W, Y, F, P) are shown in black.

Because identical overall results were observed with the two NSI-defective lines, we decided to merge the quantitative data for further statistical analysis and compared them to the wild type (WT). We first analyzed the data from NTA of residues 1 and 2, which almost exclusively arose from cytosolic proteins (Fig 5C). As we observed no difference, we concluded that the *nsi* knockout did not impact cytosolic NTA. We next focused on NTA occurring downstream from residue 2, usually as a result of leader peptide excision (Fig 5D). We observed a strong decrease in this subset, which contained a majority of plastid proteins. In agreement, when focusing exclusively on plastid-localized proteins (Fig 5E), the decrease was more pronounced and indicated that decreased NTA, due to inactivation of GNAT2, occurred specifically in this organelle. In addition, when focusing on the most affected proteins among all quantified N-termini, we retrieved only nuclear-encoded plastid proteins (Fig 5F). We next checked, which of the plastid proteins were affected in their NTA yield. The six major target proteins corresponded to AT2G24820 (TIC55), AT4G24830 (ASSY), AT3G54050 (F16P1), AT1G16080 (unknown protein), AT4G27440 (PORB), and AT1G03630 (PORC; Table 2 and Dataset EV4).

We next investigated whether GNAT2 has a broad activity on plastid proteins as suggested from our aforementioned GAP assay. The N-terminal sequence of the proteins featuring decreasing NTA yield in *nsi* mutant lines, in comparison with those that were unchanged, is displayed in Fig 5G. In agreement with the GAP analysis (Fig 5C, left-hand side), we also did not observe a specificity for the N-terminal residue (position 1) for GNAT2 in the *in vivo* data. In addition, the iceLogo analysis revealed that GNAT2 prefers small residues at downstream positions. These data are in agreement with the specificity derived from the GAP assay reported in Fig 4B. An interesting observation was made for fructose-1,6-bisphospahtase (F16P1), which was identified with two distinct neo-N-termini at positions Ala60 and Val61. This is not an unusual situation, since multiple N-termini have been observed for plastid-imported proteins before (Bienvenut *et al*, 2012; Rowland *et al*, 2015). In both cases, a strongly reduced acetylation of both N-termini was observed (44 → < 1% and 87 → 28%, respectively), although Val61 was less strongly affected than Ala60. This indicates that other plastid GNATs are able to acetylate F16P1 (at least at one of the two N-termini) in the absence of GNAT2. Furthermore, the data suggest that some substrate specificity, possibly arising from long-distance contacts, might exist between GNAT2 and protein targets allowing proteins displaying sequence similarity but with distinct N-termini to be modified by GNAT2. This could also explain why PORB and PORC are strongly affected while displaying rather distinct N-termini. Because the impact on F16P1 is incomplete with Val61 but complete with Ala60, this indicates that several plastid GNATs may contribute to NTA of the same site, leading to a compensation and partial acetylation if one is absent. Finally, we noticed that most affected KAed and NTAed targets of GNAT2 are of different identity, suggesting that recognition modes differ for either acetylation mode (Table 2).

Altogether, these data demonstrate that (i) GNAT2 has a predominant impact on plastid-imported proteins, therefore acting as a post-translational acetylase, (ii) GNAT2 displays both KA and NTA activity *in planta*, and (iii) about 5% (9/143 with NTA > 10%) of the plastid proteins rely on GNAT2 to achieve their own NTA. This number is consistent with a family of eight members, each

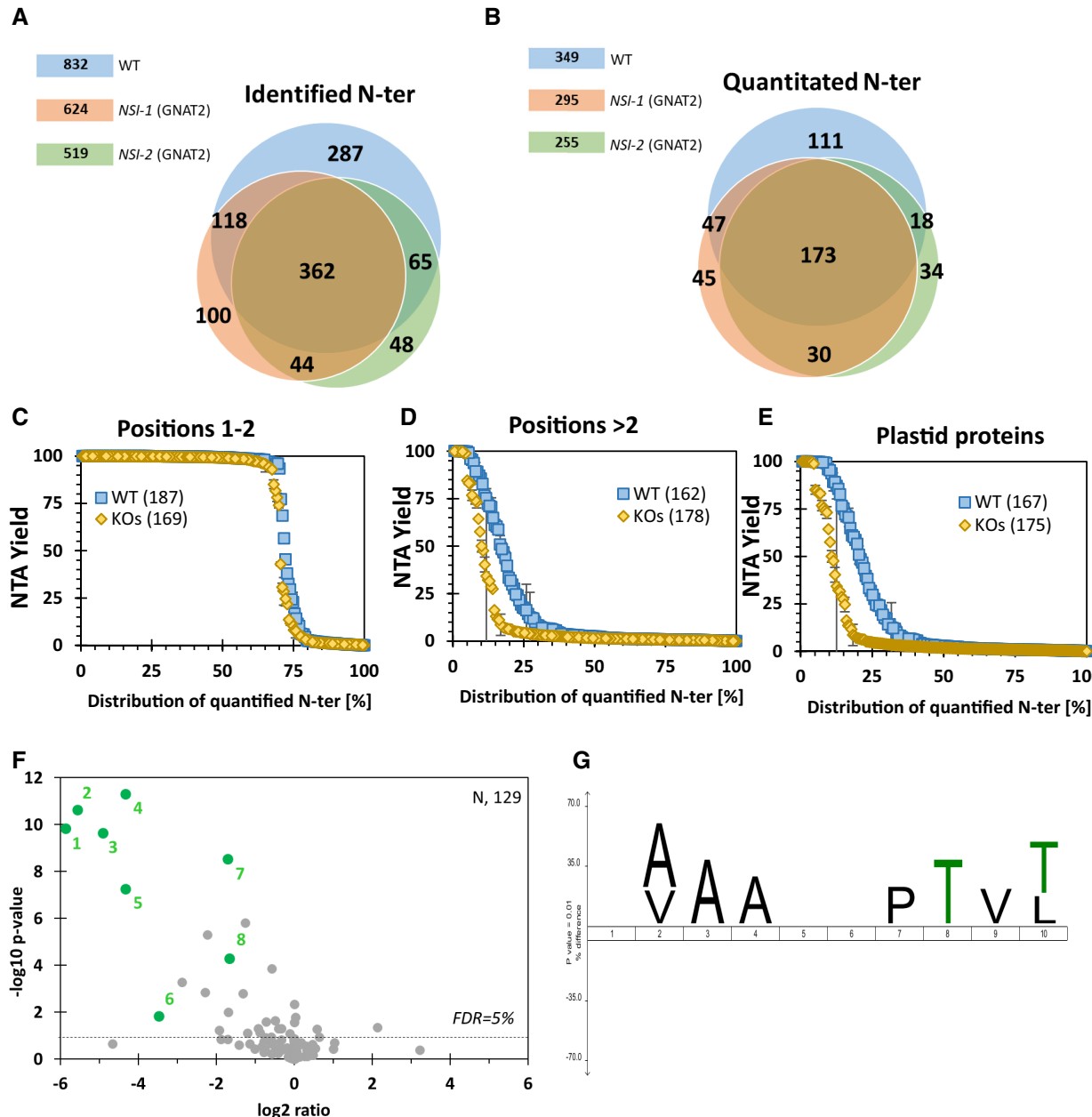

**Figure 5.** Inactivation of GNAT2 (NSI) unveils dual KA and NTA activity in planta and distinct targets for both acetylations.

A   Venn diagram representing the overlap between the N-termini sets identified in wild-type and *nsi* mutant lines. N-termini from four replicates of *Arabidopsis* wild-type and two independent *nsi* knockout lines (*nsi-1* and *nsi-2*) were retrieved as previously reported (Koskela *et al*, 2018) and compared. More than 300 N-termini could be retrieved in all samples (Linster *et al*, 2015; Huber *et al*, 2020)

B   Venn diagram of quantified NTAed proteins. Half of the retrieved N-termini could be quantified and 173 were common to all samples.

C   Comparison of NTA yield of retrieved N-termini of proteins starting at position 1 or 2. The majority of these proteins, corresponding mostly to cytosolic components, undergoing or not to N-terminal methionine excision, were not affected by inactivation of GNAT2. For statistical analyses, *nsi-1* and *nsi-2* were pooled and compared to the wild type. Two independent technical replicates of four biological replicates for each of the WT, *nsi-1*, and *nsi-2* samples were analyzed. Error bars are ± SD.

D   Comparison of NTA yield of retrieved N-termini of proteins starting at positions > 2. Clear alteration of NTA yield was observed in *nsi* mutant lines in the pool of nuclear-encoded plastid proteins. *nsi-1* and *nsi-2* samples were treated as in (C). Error bars are ± SD (see details of sampling in panel C).

E   Comparison of NTA yields of retrieved N-termini in plastid proteins. Similar variation as in panel (D) was observed when NTA of only plastid proteins was analyzed. Error bars are ± SD (see details of sampling in panel C).

F   Volcano plot representing NTA analyses of *nsi* knockout lines (treated together) and wild type. For this analysis, the *P*-value was calculated using Excel's two-tailed t-test function, for two-sample with equal variance. The most impacted proteins are shown in green. See Table 2 for correspondence. N is related to the number of quantified N-termini.

G   IceLogo representation (Colaert *et al*, 2009) of the protein N-termini with modified NTA yield vs proteins with unmodified NTA yield. The color symbol associated with each residue is detailed in the legend to Fig 4B. Black is aliphatic; green is small hydrophilic.

**Table 2. Comparison of the most affected NTAed and KAed substrates resulting from GNAT2 inactivation.**

| Spot # | Accession ARAPORT-11 | Subcellular localization | Protein ID | Acetylation position | NTA Ratio (KO/WT) | KA Ratio (KO/WT) | Protein expression |
|---|---|---|---|---|---|---|---|
| 1 | AT2G24820 | Plastid | TIC55 | 51 (Nt) | 0.02 | n.i. | + |
| 2 | AT3G54050 | Plastid | F16P1 | 60 (Nt); 323 (K) | 0.02 | 0.97 | + |
| 3 | AT1G16080 | Plastid | Q9S9M7 | 45 (Nt); 275 (K) | 0.03 | 1.31 | + |
| 4 | AT4G27440 | Plastid | PORB | 68 (Nt) | 0.05 | n.i. | + |
| 5 | AT1G03630 | Plastid | PORC | 69 (Nt); 334 (K) | 0.05 | 1.08 | + |
| 6 | AT4G34290 | Plastid | Q9SYZ4 | 50 (Nt) | 0.09 | n.i. | n.i. |
| 7 | AT4G24830 | Plastid | ASSY | 75 (Nt) | 0.31 | n.i. | + |
| 8 | AT3G54050 | Plastid | F16P1 | 61 (Nt); 323 (K) | 0.32 | 0.97 | + |
| 1* | AT1G01790 | Plastid | KEA1 | 168 (K) | n.i. | 0.02 | + |
| 2* | AT2G05310 | Plastid | Q9SJ31 | 62 (K) | n.i. | 0.20 | + |
| 3* | AT5G01600 | Plastid | FRI1 | 134 (K) | n.i. | 0.65 | + |
| * | AT1G06680 | Plastid | PSBP1 | 41 (Nt); 88 (K) | 1.17 | 0.08 | + |
| * | AT2G34430 | Plastid | Q39142 | 40 (K) | n.i. | 0.81 | + |
| * | AT1G52230 | Plastid | PSAH2 | 138 (K) | n.i. | 0.82 | + |

Proteins with a significant decrease in acetylation according to a FDR < 5% in the NTA experiment (1–8, see also Fig 5F) or in the KA experiment (*, see Fig 2 and Supplemental Dataset 1 of Koskela *et al*, 2018); *n.i.*: protein not identified in the experiment; + : protein expression found stable in the global quantitation experiment.

contributing to 10–20% of the overall plastid NTA capacity and with likely partial redundancy.

# Discussion

Acetylations are among the most abundant and essential protein modifications, which can occur on the amino group of N-termini and internal K residues of proteins, respectively. Both acetylation mechanisms involve the transfer of an acetyl moiety from acetyl-coenzyme A (Ac-CoA) to the Nα- or Nε-lysine amino group of an acceptor amino acid by the action of NATs or KATs, respectively. In a given proteoform, there is only one Nα and an average of 32 Nε-K acceptors (preprint: Agoni, 2015; Kozlowski, 2016). Most of the published knowledge on NTA arises from yeast and human studies on the cytosolic NAT isoforms. The cytosol of these organisms contains an increasing number of NATs often assembled in complexes (seven as to the year 2020, named NatA/B/C/D/E/F/NAA80), which either act co- or post-translationally. All cytosolic NATs characterized so far display different, narrow substrate specificities. For instance, the NatA complex modifies all proteins having lost their iMet as a result of the essential N-terminal methionine excision process (NME) ensured by methionine aminopeptidases (MetAP enzymes). MetAPs are able to cleave iMet only if amino acid at position two displays a short lateral chain (i.e., A, G, S, and T), representing almost half of the cytosol isoforms (Giglione *et al*, 2015; Aksnes *et al*, 2016; Breiman *et al*, 2016). In contrast, NatB and NatC modify proteins retaining their iMet, and their specificity depends on the nature of the amino acid at position P2′. The other cytosolic isoforms acetylate a well-defined set of proteins (Drazic *et al*, 2018). The plant cytosolic NTA machinery arises from the characterization of N-termini from many photosynthetic organisms (Martinez *et al*, 2008; Zybailov *et al*, 2008; Liu *et al*, 2013; Linster & Wirtz, 2018). Characterization of *A. thaliana* NatA shows conservation with metazoan NATs (Linster *et al*, 2015; Xu *et al*, 2015). Previous studies tend to favor separated enzymes for the modification of the protein N-termini and K residues (Liszczak *et al*, 2011; Magin *et al*, 2016).

Although the number of experimentally characterized NTAed and/or KAed proteins is continuously increasing, many features of the enzymes that catalyze these modifications are still unknown, especially in prokaryotic cells and organelles such as mitochondria and chloroplasts. Chloroplasts evolved from engulfed prokaryotes, most likely ancient cyanobacteria that once lived as independent organisms, and therefore these organelles resemble bacteria in certain aspects. Nonetheless, the new cellular resident evolved acquiring unique features with a simultaneous reduction of the genome size due to relocation of genes to the nucleus. In case of higher plant plastids, this genome reduction resulted in only about one hundred protein-coding genes remaining in the plastome. Recently, an unexpected feature of chloroplasts has been highlighted and concerns both KA and NTA. Although these modifications affect only a minority (between three and 10%) of soluble proteins in bacteria, KA and particularly NTA have been found on more than 30% of plastid proteins in *Arabidopsis*, including those encoded by the plastid genome (Finkemeier *et al*, 2011; Wu *et al*, 2011; Bienvenut *et al*, 2012; Rowland *et al*, 2015; Hartl *et al*, 2017).

Despite the recent identification of NAA70/GNAT4 as a plastidial NAT enzyme, and NSI/GNAT2 as a KAT enzyme (Dinh *et al*, 2015; Koskela *et al*, 2018), it was unclear so far if other KAT and NAT enzymes were present in plastids, especially since only a limited number of protein substrates were revealed for NSI/GNAT2 (Koskela *et al*, 2018). In the case of KA, acetylation might also occur non-enzymatically in the chloroplast stroma during active photosynthesis, when the pH is rising to about eight (Hohner *et al*, 2016). Hence, the question to what extent KA is enzymatically or non-enzymatically controlled in chloroplasts is still unanswered, especially since it is expected to occur mainly non-enzymatically in

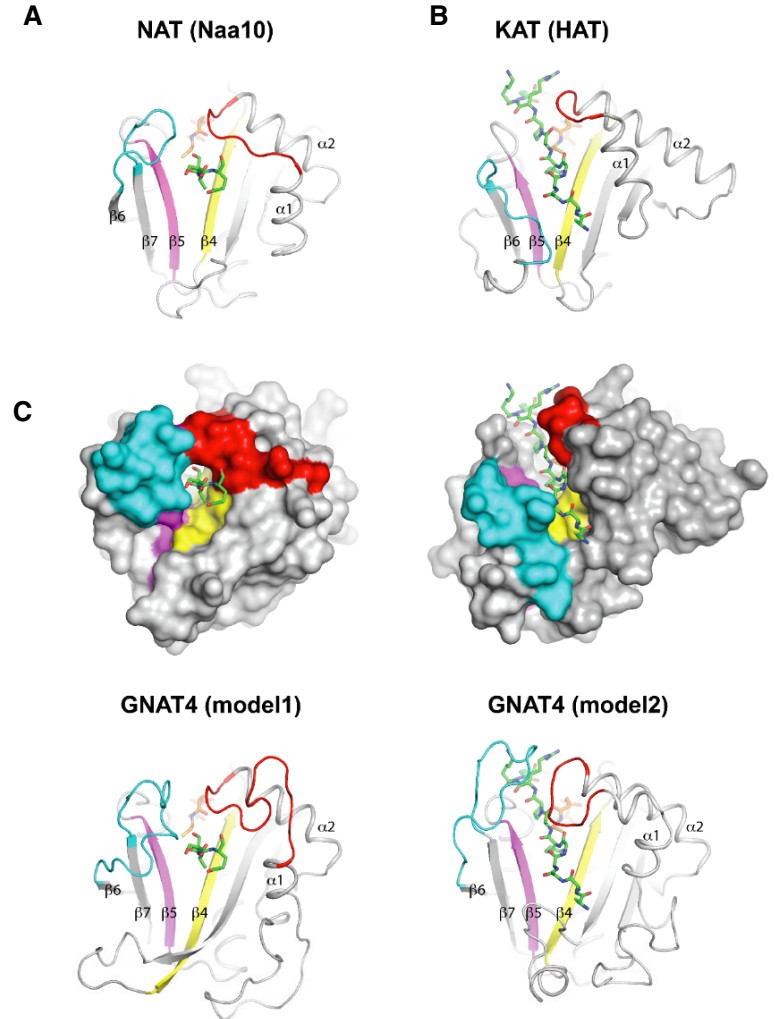

**Figure 6. NAT and KAT activities of plastid GNATs suggest a dual conformation of both α1α2 and β6β7 loops.**

Specific NAT or KAT activity of GNAT members was previously suggested to be dependent on the GNAT-fold (Magin *et al*, 2016).

A   Crystal structure of the catalytic subunit of NatA from yeast complexed with a bisubstrate inhibitor (PDB code 4KVM). The interaction of the α1α2 and β6β7 loops restricts access to the binding site of the peptide promoting *N*-α acetylation of substrate peptide.

B   Crystal structure of the HAT domain of *Tetrahymena* GCN5 bound with both peptide ligand and CoA (PDB code 1QSN). The absence of a β7 strand and the different C terminus creates an accessible groove at the surface of the HAT protein promoting internal *N*-ε lysine acetylation of the substrate peptide. The main chain (top) and solvent accessibility (bottom) of both proteins are displayed as gray ribbon or gray surface, respectively. α1α2 loop, β4 strand, β5 strand are colored in red, yellow, and pink. β6β7 loop in A and β6α4 loop in B are colored in cyan. Peptides and the CoA moiety are displayed as green and orange sticks, respectively.

C   Models of GNAT4 obtained from structure homology-modeling server SWISS-model using the pdb codes of two GCN5-related *N*-Acetyltransferases (2 × 7b, left/model 1; 4H89, right/model 2) as template. The GNAT4 models show different α1α2 and β6β7 loop conformation, suggesting a mobile loop allowing GNAT4 to ensure a KA/NTA dual activity.

mitochondria (König *et al*, 2014). However, KA has recently been identified also in non-green plastids of roots (Uhrig *et al*, 2019), which provides support to the enzymatically regulated KA. The identification of the enzymes responsible for KA or NTA has been hampered by the low sequence homology shared by these proteins. All known NATs belong to the superfamily of GNATs, as do most identified KATs (Montgomery *et al*, 2015; Drazic *et al*, 2016). GNAT Prosite motifs, associated with the main eukaryotic cytosolic NATs or KATs, are generally used for the search of these enzymes in different genomes where they have not been annotated so far. However, this type of search does not allow to discriminate between

the activities of the different types of GNATs. Our parallel and multi-layered strategies in the search of putative plastid NATs or KATs revealed a common pool of proteins with a number of unique features, both at the level of conserved motifs (e.g., the Ac-CoA BD) as well as their activities. This was never before observed together in known NATs and KATs. Indeed, the majority of plastid GNATs, in contrast to their cytosolic counterparts, possess two Ac-CoA BDs, with the most conserved one located at the N-terminus of the α3-helix and displaying the following specific pattern: [RQ]xxG[LIMV][AG]xx[LIMVF][LIMV]. The second, more degenerated Ac-CoA BD, was often found upstream of the main one. In addition to

chloroplast GNATs, our investigation revealed a secondary Ac-CoA BD only on human NAA30 and *Arabidopsis* NAA40. Why the majority of plastid GNATs display a secondary Ac-CoA BD is still unknown. However, the additional observation that the different plastid GNATs separately share conserved residues, previously shown to be involved in substrate binding and specificity in all cytosolic NAA, suggests that those plastid GNATs might have specific and unique substrate specificities. Our global N-α-acetylome profiling, indeed, reveals that the substrate specificity of all plastid GNATs is more similar to that of archaeal NATs (Liszczak & Marmorstein, 2013) than to any other NAT described so far in eukaryotes or bacteria. Surprisingly, all identified GNATs display a clear dual KAT and NAT activities. The comparison between cytosolic NatA catalytic subunit (NAT) and nuclear HAT bound with their cognate peptide substrates reveals that the conformation of α1α2 and β6β7 loops may be the determinant for their different activities, as previously suggested (Magin *et al*, 2016) (Fig 6A and B). In NAA10, α1α2 and β6β7 loops interact with each other to constrain the access to the active site favoring N-terminal peptide recognition (Fig 6A). In contrast, the absence of β6β7 strands in HAT leads to a different organization of α1α2 loop and the C terminus, resulting in a large catalytic groove suitable for internal lysine-peptide recognition (Fig 6B). Interestingly, all identified GNATs show either insertions of structural elements in α1α2 loop and/or missing β6β7 loops that may explain their dual activities. Homology models of GNAT4, displaying both KAT and NAT activities, suggest that longer predicted α1α2 and β6β7 loops could adopt two different conformations compatible with either NTA or KA activities (Fig 6C). In line with this observation, it has recently been proposed that hydroxylation of W38 of human NAA10 widens the substrate gate of the enzyme enabling it to acquire a KA activity (Kang *et al*, 2018).

How can we explain the diversification of the plastid GNAT compared to the cytosolic or bacterial counterpart? We can expect that these unique proteins underwent diversification through evolution to cope with specific plastid functions. In this context, we can hypothesize that while KA affects the activity of several photosynthetic enzymes, including RuBisCO (Finkemeier *et al*, 2011; Gao *et al*, 2016), NTA might affect the protein stability and might be a part of a plastidial *N*-degron pathway (Bouchnak & van Wijk, 2019) as previously suggested (Bienvenut *et al*, 2011; Hoshiyasu *et al*, 2013), or it might be involved in other not yet uncovered functions. In addition, the different substrate specificities and in particular the

low activities observed for GNAT1 and 3 suggest that each of these proteins works in a specific subcompartment of the plastid or on specific plastid protein families. Deletion of the enzyme GNAT2 (NSI) had no effect on plant phenotype under constant light, but resulted in a strong reduction in growth when plants were subjected to fluctuating light conditions (Koskela *et al*, 2018, 2020). Intriguingly, the *nsi* plants were not able to respond to changes in light conditions by balancing the absorbed light energy between the photosystems (PS) through state transitions and the *nsi* plants were locked in state 1, referring to the association of LHCII to PSII (Koskela *et al*, 2018). Association of LHCII to PSI (state 2) occurs via interaction of the LHCII trimer with the PSI docking site comprised of PSAH/L/O in a process determined by phosphorylation status of LHCII (Lunde *et al*, 2000; Crepin & Caffarri, 2015; Longoni *et al*, 2015). In *nsi* plants, however, no defects in LHCII phosphorylation could be detected, but instead KA of PSAH as well as LHCB1.4 was decreased, which might hinder the binding of LHCII to PSI. It is also possible that loss of GNAT2/NSI may have other downstream effects on thylakoid dynamics and thereby on state transitions. In *nsi*, several chloroplast proteins showed a decrease in NTA as well. However, none of these proteins were involved in light reactions of photosynthesis, and thus, direct involvement of NTA in the regulation of light harvesting through state transitions is unlikely. Nevertheless, an interesting recent study showed extensive light-independent NTA of stroma-exposed N-terminal loops of many of the PSII-LHCII proteins. This suggests that NTA may play a role in the formation of grana stacks via PSII-LHCII supercomplex interaction through the stromal gaps (Albanese *et al*, 2020) and thus be involved in organization of thylakoid protein complexes.

Taken together, our results show that NSI/GNAT2 displays an additional NTA activity *in vivo* next to its KA activity, and that the acetylation recognition mode of GNAT2 differs between KA and NTA. Thus, GNAT enzymes may have unexpected impacts on the regulation of photosynthesis and chloroplast metabolism in general. Characterization of the novel chloroplast GNAT family has revealed a new layer of complexity in the enzymatic machinery responsible for acetylation and suggests that other eukaryotic GNATs may also have these unforeseen broader enzymatic activities. Such properties were most recently also reported for a *N*-myristoyltransferase, a GNAT involved in acyl transfer to the N-termini of proteins (Castrec *et al*, 2018; Dian *et al*, 2020; Kosciuk *et al*, 2020; Meinnel *et al*, 2020).

# Materials and Methods

## Reagents and Tools table

| Reagent/resource | Reference or source | Identifier or catalog number |
|---|---|---|
| **Experimental models** | | |
| *Escherichia coli* Rosetta™(DE3) | Merck | |
| *Escherichia coli* BL21(DE3)pLysS | Thermo Fisher | |
| *Arabidopsis thaliana* (Col-0) | | |
| *Arabidopsis thaliana* (Col-0)*nsi-1* (SALK_033944) | Koskela *et al* (2018) | |
| *Arabidopsis thaliana* (Col-0)*nsi-2* (SALK_020577) | Koskela *et al* (2018) | |

**Reagent and Tools table** (continued)

| Reagent/resource | Reference or source | Identifier or catalog number |
|---|---|---|
| *Arabidopsis thaliana* Col-0, 35S:*GNAT1-GFP* | This study | |
| *Arabidopsis thaliana* Col-0, 35S:*GNAT2-GFP* | This study | |
| *Arabidopsis thaliana* Col-0, 35S:*GNAT3-GFP* | This study | |
| *Arabidopsis thaliana* Col-0, 35S:*GNAT4-GFP* | This study | |
| *Arabidopsis thaliana* Col-0, 35S:*GNAT5-GFP* | This study | |
| *Arabidopsis thaliana* Col-0, 35S:*GNAT6-GFP* | This study | |
| *Arabidopsis thaliana* Col-0, 35S:*GNAT7-GFP* | This study | |
| *Arabidopsis thaliana* Col-0, 35S:*GNAT8-GFP* | This study | |
| *Arabidopsis thaliana* Col-0, 35S:*GNAT9-GFP* | This study | |
| *Arabidopsis thaliana* Col-0, 35S:*GNAT10-GFP* | This study | |
| **Recombinant DNA** | | |
| Oligonucleotides and sequence-based reagents | This study | Table EV3 and EV4 |
| **Antibodies** | | |
| Acetyllysine antibody, for Western blot analysis, anti-rabbit | ImmuneChem | ICP0380 |
| Acetyllysine antibody, immobilized to beaded agarose | ImmuneChem | ICP0388 |
| Secondary HRP-conjugated antibody, goat anti-rabbit | Agrisera | AS09602 |
| **Chemicals, enzymes, and other reagents** | | |
| Acetic anhydride-d6, 99 atom % D | Sigma-Aldrich | 175641 |
| Acetone, for HPLC, ≥ 99.9% | Sigma-Aldrich | 270725 |
| Acetonitrile, for HPLC, ≥ 99.9% | Sigma-Aldrich | 34851 |
| Ammonium bicarbonate, BioUltra, ≥ 99.5% (T) | Sigma-Aldrich | 09830 |
| Complete™ protease inhibitor cocktail tablets | Sigma-Aldrich | 11697498001 |
| Dimethyl sulfoxide, anhydrous, ≥ 99.9% | Sigma-Aldrich | 276855 |
| Dipotassium hydrogenophosphate ($K_2HPO_4$), ≥ 98% | Sigma-Aldrich | P9791 |
| Disodium hydrogen phosphate, water free, ≥ 99% p.a. | Roth | P030 |
| Dithiothreitol, ≥ 98% | Sigma-Aldrich | D9779 |
| EGTA, ≥ 97% | Sigma-Aldrich | E4378 |
| Formaldehyde solution, 36.5–38% in $H_2O$ | Sigma-Aldrich | F8775 |
| Formaldehyde-D2 (D, 98%; ~ 20% w/w IN $D_2O$) | Cambridge Isotope Laboratories | DLM-805-20 |
| Formic acid, ≥ 95% | Sigma-Aldrich | F0507 |
| Guanidine hydrochloride, ≥ 99% | Sigma-Aldrich | 369080 |
| Glycerol, ≥ 99% | Sigma-Aldrich | G5516 |
| HEPES, ≥ 99.5% | Sigma-Aldrich | H3375 |
| *N*-hexane, anhydrous, 95% | Sigma-Aldrich | 296090 |
| Hydrochloric acid, 37% | Sigma-Aldrich | 258148 |
| Hydroxylamine solution, 50 wt.% in $H_2O$ | Sigma-Aldrich | 438227 |
| *N*-hydroxysuccinimide, 98% | Sigma-Aldrich | 130672 |
| Iodoacetamide, ≥ 99% | Sigma-Aldrich | I6125 |
| Lysozyme, lyophilized | Roth | 8259 |
| Methanol for HPLC/MS | Fisher Scientific | M406215 |
| Magnesium chloride (MgCl2) anhydrous, ≥ 98% | Sigma-Aldrich | M8266 |
| Phenylmethanesulfonyl fluoride (PMSF), ≥ 99% (T) | Sigma-Aldrich | 78830 |
| Pierce™ 660 nm Protein Assay Reagent | Thermo Fisher | 22660 |
| Potassium chloride, ≥ 99% | Sigma-Aldrich | P9333 |
| Potassium hydroxide, 90% | Sigma-Aldrich | 484016 |

**Reagent and Tools table**  (continued)

| Reagent/resource | Reference or source | Identifier or catalog number |
|---|---|---|
| Sodium chloride, ≥ 99.5% | Sigma-Aldrich | S7653 |
| Sodium cyanotrihydridoborate, ≥ 95% | Sigma-Aldrich | 156159 |
| Sodium dihydrogen phosphate, monohydrate, ≥ 98% p.a. | Roth | K300 |
| Sodium hydroxide, ≥ 98% | Sigma-Aldrich | S8045 |
| SuperSignal™ West Dura Extended Duration ECL substrate | Thermo Fisher | 10220294 |
| Thiourea, ≥ 99% | Sigma-Aldrich | T7875 |
| Trifluoroacetic acid, ≥ 99% | Sigma-Aldrich | 302031 |
| Triton X-100 | Sigma-Aldrich | ×100 |
| Tris hydrochloride, ≥ 99% | Sigma-Aldrich | T3253 |
| Trypsin (lyophilized) from bovine or porcine pancreas, TPCK Treated, ≥ 10,000 BAEE units/mg protein | Sigma-Aldrich | T1426 |
| Urea, ≥ 99.5% p.a | Roth | 2317 |
| Water, for UHPLC, for mass spectrometry | Sigma-Aldrich | 900682 |
| ARAPORT-11 protein database (fasta file) | | |
| ClustalW | Larkin *et al*, 2007 | http://www.clustal.org/ |
| EnCOUNTer v1.0 | In-house software Bienvenut *et al* (2017b) | |
| *Escherichia coli* (strain K12) database (Proteome ID: UP000000625) | | https://www.uniprot.org/ |
| IceLogo | Colaert *et al* (2009) | https://iomics.ugent.be/icelogoserver/ |
| Mascot 2.4 | Matrix Science | |
| Mascot Distiller 2.5.1 | Matrix Science | |
| MaxQuant version 1.5.2.8 | Cox and Mann (2008), Tyanova *et al* (2016a) | http://www.maxquant.org/ |
| Mega-X | Kumar *et al* (2018) | |
| Perseus version 1.5.5.3 | Tyanova *et al* (2016b) | |
| Python 2.7 | | |
| R 3.3.1 | R Core Team (2016) | https://www.r-project.org/ |
| TargetP1.1 | Emanuelsson *et al* (2007) | |
| **Material and other consumables** | | |
| 17-cm frit-less silica emitters, 0.75-μm inner diameter | New Objective | N.A. |
| Benchtop centrifuge (able to reach 12,000 *g*) | Eppendorf | N.A. |
| ChemiDoc™ gel imaging system | Bio-Rad | N.A |
| Centrifuge 5427R | Eppendorf | N.A. |
| Centrifuge 5804R | Eppendorf | N.A. |
| Dionex UltiMate 3000 RSLC, with diode array detector | Thermo Scientific | N.A. |
| EASY-nLC 1200 system | Thermo Fisher | N.A. |
| Easy-nLC-II system | Thermo Scientific | N.A. |
| Empore™ Styrene Divinyl Benzene (SDB-RPS) Extraction Disks | Supelco | 66886-U |
| Iron beads, 3 and 5 mm diameters | N.A. | N.A. |
| Liquid nitrogen | N.A. | N.A. |
| LTQ-Orbitrap Velos mass spectrometer | Thermo Scientific | N.A. |
| Mixer mill | Retsch (or equivalent) | N.A. |
| Polysulfoethyl A column (200 × 2.1 mm, 5 μm, 200 Å) | PolyLC | 202SE0502 |
| Q Exactive HF mass spectrometer | Thermo Fisher | N.A. |

| Reagent/resource | Reference or source | Identifier or catalog number |
| --- | --- | --- |
| ReproSil-Pur 120 C18-AQ, 1.9 μm | Dr. Maisch | r119.aq.001 |
| Reverse-phase C18 analytical nano-column (75 μm × 120 mm, 3 μm) | Nikkyo Technos | NTCC-360/75-3 |
| Reverse-phase C18 nano-pre-column (75 μm × 20 mm, 3 μm) | NanoSeparation | NS-MP-11 |
| Sep-Paks C18 plus short columns | Waters | WAT020515 |
| Sep-Pak tC18 SPE cartridges (1 ml phase volume) | Waters | WAT054960 |
| Sonicator UIS250V with Sonotrode LS24d3 | Hielscher | N.A. |
| SpeedVac/concentrator, with vacuum < 1 torr | N.A. | N.A. |

## Methods and Protocols

### In silico searches for putative Arabidopsis plastid NAT and KAT genes

In the search of putative plastid NAT and KAT genes, we combined two parallel strategies. The first one aimed to identify putative chloroplast (Cp) NATs based on their GNAT profile combined with the subcellular localization. In this context, a first pre-list of candidates was obtained using the GNAT PROSITE motif (Sigrist et al, 2013) (i.e., PS51186, associated with the main eukaryotes cytosolic NAT catalytic subunits such as NAA10, NAA20, NAA30, NAA40, NAA50, and NAA60, ssArd1but also EcRimI) against the A. thaliana open reading frames. This step provides 49 potential GNAT candidates (101 including all possible gene translated versions). This initial list was submitted to TargetP (Emanuelsson et al, 2007) to determine the subcellular localization of the candidates. Since this tool encounters frequent erroneous predictions between mitochondrial (Mt) and Cp localizations, both Cp and Mt predicted candidates were considered. Thus, 16 gene products (33 possible translated products) remained at this stage. Few of these candidates appear to be HATs, nuclear transcription-associated proteins or involved in amino acid synthesis in the chloroplast. Finally, a list of 10 potential Cp NATs (seven predicted at the Cp and three at the Mt) was retained.

The second strategy aiming to identify putative Cp KATs involved the analysis of the GNAT-related acetyltransferases by searching the Arabidopsis genome for proteins containing the acetyltransferase domain 1 (GNAT, PF00583), which includes the classical yeast nuclear general control non-repressed 5 (GCN5) HAT (Brownell et al, 1996) that is also present in the Arabidopsis genome (At3g54610). This search resulted in a list of 35 Arabidopsis candidate proteins, of which 10 contained a putative organellar targeting peptide according to the Suba2 database (Heazlewood et al, 2007). Both strategies converged to the same list of genes (Table EV1).

### GFP fusion and plant transformation

The open reading frames of all GNATs were amplified without stop codon from Arabidopsis (Col-0) cDNA using the primers listed in Table EV3. In case of GNAT3, 8, and 9, entry clones were generated with the pENTR™/D-TOPO™ kit (Thermo Fisher). For entering the destination vector system pK7FWG2 (Karimi et al, 2007), LR recombination reactions were performed. GNAT1, 2, 4, 5, 6, 7, and 10 coding sequences were cloned into the pGWR8 vector (Rozhon et al, 2010) before being transferred to pK7FWG2 by type II endonuclease restriction and subsequent DNA ligation. Vector constructs were

verified by sequencing and used for transient or stable expression in Arabidopsis (Col-0) plants. Agrobacterium-mediated, stable transformation was performed as previously described (Clough & Bent, 1998).

### Protoplast isolation, protoplast transformation, and confocal laser scanning microscopy

Arabidopsis wild-type Col-0 as well as stable overexpressor plants were grown for six weeks in 8-h light/16-h darkness conditions prior to the preparation of leaf protoplasts. Protoplast isolation was performed according to the tape-sandwich method (Wu et al, 2009). For transient transformation, wild-type protoplast suspensions were processed by the polyethylene glycol method (Damm et al, 1989; Frank et al, 2008). Transfected protoplasts were incubated in buffer W1 (4 mM MES-KOH pH 5.7, 0.5 M mannitol, 20 mM KCl) for 8–24 h under constant agitation and application of low light intensity (25 μmol·m$^{-2}$·s$^{-1}$ photosynthetic photon flux) prior to imaging. Confocal laser scanning microscopy (CLSM) was performed by using a Leica SP5 imaging systems (Leica Microsystems) in combination with the water immersion objective lens HCX PL APO lambda blue 63.0 × 1.20 WATER UV and an argon laser source for eGFP detection. eGFP fluorescence was measured at 490–520 nm by applying an excitation wavelength of 488 nm. In case of GNAT6-GFP localization, additional CLSM approaches were performed (Fig EV3) and protoplasts were co-transformed with an inner nucleus membrane (INM) marker coding plasmid (SUN1-OFP, Rips et al, 2017) or treated with reagents to enable nucleus (Hoechst 33342, Thermo Fisher) or mitochondria (MitoTracker Orange CMTMRos, Invitrogen) detection. In these cases, fluorescence signals were recorded at excitation/emission wavelengths of 561/590–620 (OFP), 405/440–480 (Hoechst 33342), or 514/560–590 (MitoTracker Orange CMTMRos).

### Cloning and expression of GNATs in E. coli

GNAT open reading frames were amplified from Arabidopsis (Col-0) cDNA using the following primers (Table EV4), which excluded the coding region for the transit peptide but included the stop codon of the CDS. The PCR product was cloned in pETM-41 that allows expression and purification of the recombinant GNAT protein N-terminally fused to an His$_6$_maltose-binding protein (MBP) protein construct. As control, a pETM-41 empty-vector construct was designed enabling the expression of His$_6$-MBP only. The resulting plasmids were verified by sequencing. For heterologous overexpression, E. coli BL21(DE3)pLysS (Thermo Fisher) or E. coli Rosetta™(DE3) cells (Merck) were transformed with the expression plasmid constructs. BL21(DE3)pLysS expression cultures were

incubated at 37°C until a cell density of $OD_{600} = 0.6$ was reached, followed by the addition of the deacetylase inhibitor nicotinamide at a concentration of 50 mM (in case of lysine acetylation analyses) as well as of 1 mM IPTG. During protein expression, cell cultures were incubated overnight at room temperature at 180 rpm. Transformed Rosetta™(DE3) cells were cultivated at 37°C to a cell density of $OD_{600} = 0.6$ and supplemented with 50 mM nicotinamide (in case of lysine acetylation analyses). After induction of protein expression by 1 mM IPTG, expression cultures were incubated for 3 h at 37°C and 180 rpm. The cells were harvested by centrifugation (15 min, 4,000 $g$) and pellets frozen at −80°C.

### Western blot analysis

Protein extracts of *E. coli* were separated on 12% SDS–polyacrylamide gels. Gels were either stained with Coomassie dye (3% (w/v) Coomassie G-250, 10% (v/v) ethanol, 2% (v/v) orthophosphoric acid, 190 mM ammonium sulfate) or used for blotting of proteins onto nitrocellulose membrane. Protein lysine acetylation was monitored by using an anti-acetyl-lysine primary antibody (anti-rabbit, ImmuneChem), which was incubated on the membrane overnight at 4°C. The secondary HRP-conjugated antibody (goat anti-rabbit IgG, Agrisera) was applied in a 1:10,000 dilution and detected by the ChemiDoc™ gel imaging system (Bio-Rad) by using the SuperSignal™ West Dura Extended Duration ECL substrate (Thermo Fisher).

### Purification of recombinant GNAT2 and GNAT10 proteins

The His$_6$-MBP-GNAT protein constructs were expressed in *E. coli* BL21(DE3)pLysS as described above. Afterward, cells were harvested, resuspended in buffer (50 mM Tris–HCl, pH 8, 500 mM NaCl, 5 mM $MgCl_2$, protease inhibitor cocktail [Sigma-Aldrich]), and disrupted using a French Press. The cell extract was then complemented with 5 mM DTT and 50 units of DNAse (Roche), before being loaded on a Protino Ni-NTA affinity chromatography matrix (Macherey-Nagel). His$_6$-MBP-GNAT2 or His$_6$-MBP-GNAT10 was eluted with 500 mM imidazole and desalted by gel filtration using PD-10 columns (GE Healthcare). For storage, the protein preparations were buffered in 100 mM Tris–HCl (pH 8). Protein concentration was determined with the Pierce™ 660 nm Assay Reagent (Thermo Fisher).

### Solid-phase peptide synthesis

Amino acid derivatives for solid-phase peptide synthesis (SPPS) were purchased from GL Biochem (Shanghai, China), except Fmoc-Ahx-OH, Fmoc-D-Arg-OH, and Fmoc-Lys(Dnp)-OH, which were bought from IRIS Biotech (Marktredwitz, Germany), and Fmoc-Lys(Ac)-OH which was obtained from Bachem (Bubendorf, Switzerland). HATU was bought from Fluorochem (Hadfield, UK). TentaGel S RAM resin was obtained from Rapp Polymere (Tübingen, Germany). Other chemicals were purchased from Sigma-Aldrich (Steinheim, Germany), Merck (Darmstadt, Germany), and Carl Roth (Karlsruhe, Germany). Solvents were obtained from J. T. Baker (Deventer, the Netherlands), VWR (Leuven, Belgium), Fisher Scientific (Loughborough, UK), Bio-solve (Valkenswaard, the Netherlands), and Th. Geyer (Renningen, Germany).

Peptides were synthesized by SPPS on a ResPep SL synthesizer (Intavis, Cologne, Germany) applying the Fmoc/*t*Bu strategy. The scale was 2 μmol on TentaGel S RAM resin (capacity: 0.23 mmol·g$^{-1}$)

in 96-well plates. Amino acid side chains were protected as follows: D-Arg(Pbf), Gln(Trt), Lys(Boc), Ser(O*t*Bu), Thr(*t*Bu).

Coupling reactions of amino acid building blocks (5.25 eq) were performed with HATU (2-(7-aza-1*H*-benzotriazole-1-yl)-1,1,3,3-tetramethyluronium hexafluoro-phosphate) (4 eq) as activator and NMM (*N*-methylmorpholine) (10 eq) as base in DMF twice for 20 min. After each cycle unreacted, amino groups were capped by treatment with a solution of $Ac_2O$ (5%) and 2,6-lutidine (6%) in DMF for 5 min. The Fmoc group was deprotected with piperidine (20% in DMF, 2× for a total of 15 min). Peptides with free alpha-amino group were Fmoc-deprotected after the last capping step and peptides with acetylated N-termini were Fmoc-deprotected before the final capping step. The resin was washed with DMF between each step.

Peptides were cleaved off the resin with a solution containing TFA, water, phenol, thioanisole, and 1,2-ethanedithiol (82.5:5:5:5:2.5, 600 μl per well) for 3 h in total. The solution was added in portions (1 × 200 μl and 3 × 100 μl) to each well, and the resin was incubated for 30 min after each addition, except the last one, after which it was incubated for 1.5 h. The resin was rinsed with additional cleavage solution (100 μl) and the solutions were then concentrated by evaporation under ambient conditions in a fume hood overnight. Cleaved peptides were precipitated in cold $Et_2O$ (700 μl per well), centrifuged (2,500 $g$, 20 min, −4°C), washed with additional $Et_2O$ (3×), dissolved in water/MeCN, and lyophilized. Peptides were analyzed by liquid chromatography–mass spectrometry (LC-MS), which was performed using a Shimadzu LC-MS 2020 device (Kyoto, Japan) equipped with a Kinetex 2.6 μm C18 100 Å (100 × 2.1 mm) column (Phenomenex, Aschaffenburg, Germany). Samples were prepared with LC-MS solvents A (0.1% formic acid in water) and B (80% MeCN, 0.1% formic acid in water). The analytical gradient was 5–95% B in 12.75 min with a flow rate of 0.2 ml·min$^{-1}$. Absorption was detected at 218 nm. The ESI-MS was operated in positive mode.

### Acetyltransferase activity assay

To investigate the acetyltransferase activities of the recombinant His$_6$-MBP-GNAT2 and His$_6$-MBP-GNAT10 proteins, a HPLC-based peptide assay was used as previously reported with some modifications (Koskela *et al*, 2018). The recombinant protein (5 μM) was incubated with the KAT peptide substrate (free ε-lysine, 50 μM) or a variety of NAT peptide substrates (50 μM), with the following sequences: x-A-Q-G-A-K(ac/NH$_2$)-A-A-K(Dnp)-Ahx-r-r-r-NH$_2$, with x = free or acetylated α-M, α-A, α-G, α-S, α-T, α-V, or α-L, Ahx = 6-aminohexanoic acid as a spacer. The absorbance of the dinitrophenyl group (Dnp) of the peptides was recorded at 340 nm. The reaction buffer contained 150 mM HEPES (pH 8) and 30 mM KCl. The reaction was started by addition of 100 μM acetyl-CoA for 45 min in a thermoblock at 30°C. Twenty microliter samples were collected at indicated time points, and the reaction was stopped by addition of 180 μl trifluoroacetic acid (TFA, final concentration 2% (v/v)). For analysis of reaction products, a reversed-phase HPLC chromatograph (Shimadzu) equipped with CBM-20A controller, two LC-20AD pumps, a DGU-20A degasser, an SPD-20A detector, and an SIL-20AC autosampler was used. The separation of peptides was performed on a C18 Hypersil GOLD column (4.6 mm × 250 mm, 5-μm particle size; Thermo Fisher Scientific). A gradient program was set up consisting of solvent A (0.1% TFA (v/v) in distilled water)

and solvent B (95% acetonitrile and 0.1% TFA (v/v) in distilled water), which were applied at a flow rate of 1.0 ml·min$^{-1}$ as follows: 0–1 min, 5% B; 1–20 min, linear 5–100% B; 20–25 min, 100% B; 25–25.5 min, 100–5% B; 25.5–30 min, 5% B. Samples (100 µl) were injected, and the elution of reaction products was followed at detection wavelengths of 218 nm (peptide backbone) and 340 nm (Dnp). Enzymatic activity was calculated based on the peak area values that corresponded to the KAT/NAT substrates (elution in between 13.9 and 14.05 min) and the acetylated products (elution in between 14.5 and 14.65 min).

### Protein extraction, peptide dimethyl labeling, and K-acetylated peptide enrichment

*Escherichia coli* cell pellets were resuspended in 2 ml of 50 mM sodium phosphate buffer pH 8.0, 300 mM NaCl. Cells were disrupted by three rounds of sonication (15 s per treatment with output of 70%), followed by addition of 10 units of lysozyme. Protein extracts were cleared by centrifugation (18,000 $g$, 4°C, 40 min), and proteins of the supernatant were precipitated by addition of 5 ml 100% (v/v) ice-cold acetone, incubation at −20°C for 2 h, and additional centrifugation (15 min, 14,000 $g$, 4°C). The protein pellets were dissolved in 500 µl of 6 M urea, 2 M thiourea, 10 mM HEPES, and protein concentration was determined using Pierce™ 660 nm Protein Assay Reagent (Thermo Fisher). Per sample, 5 mg of protein was further processed and diluted with 50 mM ammonium bicarbonate to adjust a final urea concentration of 2 M maximum. Proteins were digested by applying MS-grade trypsin (Serva) in a 1:100 ratio and incubation overnight at 37°C.

Digested peptides were dimethyl-labeled on C18 Sep-Pak plus short columns (Waters) as described previously (Lassowskat *et al*, 2017). Labeled peptide samples of GNAT overexpression cultures were combined in equal amounts with the corresponding control samples of labeled peptides prepared from *E. coli* cells expressing His$_6$-MBP only. Biological duplicates were measured for each sample, and a label swap was introduced (light label: dimethyl mass shift +28.0313 Da, heavy label: dimethyl mass shift +32.0564 Da). The combined peptide samples were dried in a vacuum centrifuge and resuspended in 1 ml TBS buffer (50 mM Tris–HCl pH 7.6, 150 mM NaCl). Fifteen microgram peptide was stored for whole-proteome analysis, while about 4 mg peptide was used for enrichment of lysine-acetylated peptide sites by loading the samples on agarose beads coupled to anti-acetyl-lysine antibody (Hartl *et al*, 2015). After immunoprecipitation, the enriched peptides were eluted, desalted, and fractionated in three steps by performing SDB-RPS Stage-Tipping (Kulak *et al*, 2014). At the same time, the samples stored for whole-proteome analysis were desalted and fractionated as well. The solvent was removed by vacuum centrifugation, and the dried pellets were stored at −20°C.

### LC-MS/MS data acquisition for K-acetylation

Peptide pellets were redissolved in 8 µl of 2% ACN, 0.1% TFA for LC-MS/MS analysis. For whole-proteome analyses, a peptide concentration of 0.1 mg·ml$^{-1}$ was adjusted with 2% ACN, 0.1% TFA and 0.5 µg of peptides was loaded. Samples enriched for lysine-acetylated peptides (acetylome) were loaded entirely. Samples were analyzed using an EASY-nLC 1200 (Thermo Fisher) coupled to a Q Exactive HF mass spectrometer (Thermo Fisher).

Peptides were separated on 17-cm frit-less silica emitters (New Objective, 0.75 µm inner diameter), packed in-house with reversed-phase ReproSil-Pur C18 AQ 1.9-µm resin (Dr. Maisch). The column was kept in a column oven at 50°C. Following parameters were used in whole-proteome analysis, parameters for acetylome analysis are stated in brackets; if not stated separately, parameters were identical. Peptides were eluted for 115 (68) min using a segmented linear gradient of 0–98% solvent B (solvent A 0% ACN, 0.5% FA; solvent B 80% ACN, 0.5% FA) at a flow rate of 300 (250) nl·min$^{-1}$. Mass spectra were acquired in data-dependent acquisition mode with a TOP15 method. MS spectra were acquired in the Orbitrap analyzer with a mass range of 300–1,759 $m/z$ at a resolution of 60,000 (120,000) FWHM, maximum IT of 55 ms, and a target value of $3 \times 10^6$ ions. Precursors were selected with an isolation window of 1.3 (1.2) $m/z$. HCD fragmentation was performed at a normalized collision energy of 25. MS/MS spectra were acquired with a target value of $10^5$ ($5 \times 10^4$) ions at a resolution of 15,000 FWHM, maximum IT of 55 (150) ms, and a fixed first mass of $m/z$ 100. Peptides with a charge of +1, > 6, or with unassigned charge state were excluded from fragmentation for MS2, dynamic exclusion for 30 s prevented repeated selection of precursors.

### Data analysis for K-acetylation

Raw data were processed using the MaxQuant software (version 1.5.2.8, http://www.maxquant.org/) (Cox & Mann, 2008; Tyanova *et al*, 2016a). MS/MS spectra were searched against the Uniprot *E. coli* (strain K12) database (Proteome ID: UP000000625) including the sequences of all His$_6$-MBP-GNAT proteins. Sequences of 248 common contaminant proteins and decoy sequences were automatically added during the search. Trypsin specificity was required and a maximum of two (proteome) or four missed cleavages (acetylome) were allowed. Minimal peptide length was set to seven amino acids. Carbamidomethylation of cysteine residues was set as fixed, oxidation of methionine, and protein N-terminal acetylation as variable modifications. Acetylation of lysines was set as variable modification only for the acetylome analyses. Light and medium dimethylation of lysines and peptide N-termini was set as labels. Peptide-spectrum-matches and proteins were retained if they were below a false discovery rate of 1%, modified peptides were additionally filtered for a score ≥ 35 and a delta score of ≥ 6 to remove low-quality identifications. Match between runs was enabled. Downstream data analysis was performed using Perseus version 1.5.5.3 (Tyanova *et al*, 2016b). For proteome (protein groups table) and acetylome (modification specific table), reverse hits and contaminants were removed, the normalized site ratios were log2-transformed, and label-swapped samples inverted. Plotting of the raw and the normalized site ratios confirmed that the automatic normalization procedure of MaxQuant worked reliably and normalized site ratios was used for all further analyses. For quantitative analyses, the acetylome sites were filtered for a localization probability of ≥ 0.75. Technical replicates were averaged and sites as well as protein groups displaying less than two ratios were removed.

### *Arabidopsis* material for N-terminomics analysis

*Arabidopsis thaliana* wild-type (Col-0) and two independent *gnat2* mutant lines (*nsi-1*: SALK_033944 and *nsi-2*: SALK_020577) plants

were grown as previously reported (Koskela *et al*, 2018). Four-eight distinct rosettes for each plant line were collected in the middle of the 8-h light period and pooled for one biological replicate. Altogether, four distinct biological replicates per each plant line were used to characterize and quantify NTA following the protocol described below.

### Identification and quantification of N-terminal protein acetylation

The following protocol was applied to perform both the GAP assay on *E. coli*, and the comparison between *Arabidopsis* WT and *gnat2 (nsi)* knockout mutants. The only difference will be the lysis method, depending on the biological source material. Note that all the solutions and buffers were freshly prepared. The only solution that needed to be prepared in advance was the $d_3$-*N*-acetoxysuccinimide, used for the labeling of primary amines.

### $D_3$-N-acetoxysuccinimide preparation

1   Three hundred seventy-three milligram of *N*-hydroxysuccinimide and 1 g of acetic anhydride were mixed together.
2   The tube was heated to 35–40°C to ensure complete solubilization, then placed at room temperature, under slight agitation (200 rpm) for 12–15 h.
3   The crystals were collected and the excess of solvent removed using clean filter paper.
4   The crystals were washed with 200 µl of anhydrous hexane, and clean filter paper was used to dry it. This step was repeated at least once.
5   The crystals were collected in clean PCR tubes and stored at −20°C.

### Protein extraction

1   Same pellets of *E. coli* expressing or not the different plastid GNATs used for KA were resuspended in 1 ml of lysis buffer A (50 mM HEPES/NaOH pH 7.2; 1.5 mM MgCl$_2$; 1 mM EGTA; 10% glycerol; 1% Triton X-100; 2 mM PMSF; 1 protease inhibitor tablet per 50 ml) and sonicated on ice.
2   For the N-terminal analysis of the *A. thaliana* samples, the leaflets were transferred into clean 2-ml Eppendorf tubes, and then, two iron beads of 3 and 5 mm diameter were added. Samples were flash-frozen using liquid nitrogen then grinded by using a mixer mill at 30 Hz for 30 s repeated twice. One milliliter of freshly prepared lysis buffer A was added to the powdered samples.
3   Lysates were then incubated for 1 h at 4°C, under constant agitation followed by centrifugation of the samples at 12,000 *g* at 4°C for 30 min. Supernatant was collected and protein concentration determined using Bradford or other relevant techniques.

### Sample preparation for N-terminal acetylation analysis

The GAP assay for each GNAT protein was performed as previously reported (Bienvenut *et al*, 2017a,b) and follows:

1   One milligram of total proteins was precipitated by adding four volumes of cold acetone, stirred vigorously, and placed at −20°C for at least 2 h or overnight. Samples were centrifuged at 17,000 *g*, −10°C or lower for 30 min. Supernatants were discarded, ensuring the removal as much acetone as possible.

2   The protein pellets were solubilized in 200 µl freshly prepared denaturation solution (6 M guanidine hydrochloride, 4 mM DL-dithiothreitol, and 50 mM Tris buffered with HCl at pH 8) for 15 min at 95°C.
3   After the samples were cooled down, sulfhydryl groups of the cysteines were blocked adding iodoacetamide (50 mM final concentration) and incubating the samples for 1 h at room temperature in the dark.
4   After acetone cold precipitation (1 ml) at −20°C for at least 2 h or overnight, the samples were centrifuged at 17,000 *g*, −10°C or lower for 30 min and air dry to remove as much acetone as possible. Then, the pellets were resuspended in 200 µl of phosphate buffer (50 mM KH$_2$PO$_4$/KOH, pH 7.5) and further treated with 15 µl of labeling solution (D$_3$-*N*-acetoxysuccinimide 2 M in DMSO) followed by a 90-min incubation at 30°C to favor N-terminus and ε-amino group d3-acetylation. Potential O-acetylation of S, T, and Y side chains was reversed by adding 10 µl of 50% (weight in water) of hydroxylamine and incubated for 20 min at room temperature, which also stopped the reaction.
5   The reactional mixture was acetone precipitated to remove chemical reagents, as described above, and resuspended in 300 µl of NH4HCO3 pH8. Proteins were digested by the addition of 1 µl of trypsin solution (10 µg·µl$^{-1}$ in 1 mM HCl, pH 3) and 90-min incubation at 37°C. Another 1 µl of trypsin solution was added to the mixture and incubation step repeated. The sample was acidified with formic acid to stop the reaction. After protein digestion, the peptide mixture was desalted using Sep-Pak tC$_{18}$ cartridge as recommended by the manufacturer. Eluted peptide mixture/solution was dried down and suspended in the 5 mM KH$_2$PO$_4$, 30% acetonitrile, 0.05% formic acid, and adjust at pH 3 with H$_3$PO$_4$.
6   For N-termini enrichment, peptides were separated using a strong exchange chromatography (SCX) consisted of a polysulfoethyl A column. Peptide was eluted from the SCX column using a gradient of 350 mM KCl, 5 mM KH$_2$PO$_4$, 30% acetonitrile, and 0.05% formic acid at a flow rate of 0.2 ml·min$^{-1}$. Fractions were collected every 2 min for 45 min, and the solvent was removed under vacuum until dryness.
7   In the case of *Arabidopsis* samples, fractions 2–5 and 6–11 were suspended, respectively, in 25 and 30 µl of 5% acetonitrile and 0.1% trifluoroacetic acid in water. 10 microliter of each fraction was loaded at a maximum pressure of 220 bars onto a pre-column (NS-MP-10, NanoSeparation, the Netherlands) and separated along a 55-min multistep gradient of increasing percentage of 0.1% of formic acid in acetonitrile, followed by an analytical separation using a Nikkyo Technos capillary column (NTCC-360/100-5-153, Nikkyo Technos Co., Tokyo, Japan) on an Easy NanoLC-II system at a constant flow rate of 300 nl·min$^{-1}$ coupled to an LTQ-Orbitrap™ Velos.
8   In the case of bacterial samples, fractions 2–5, 6–8, and 9–11, resuspended as *Arabidopsis* samples, were combined together and 18 µl of the resulting mixtures was loaded onto a pre-column and separated along a 120-min multistep gradient of increasing percentage of 0.1% of formic acid in acetonitrile, followed by an analytical separation using the same capillary column on the Easy NanoLC-II system at a flow rate of

300 nl·min$^{-1}$ coupled to an LTQ-Orbitrap™ Velos. Each mixture was analyzed twice.

All mass spectrometry methods used were set to acquire a survey scan (MS1) in the Orbitrap section with a mass/charge ($m/z$) range of 400–2,000 Th at 60,000 FWHM resolution, using the lock mass for internal calibration. The fragments analyses (MS2) of the 20 most intense precursor ions were performed in the LTQ section, after being subjected to collision-induced dissociation (CID) fragmentation, with a 20 s exclusion time window for the acquired precursors. When analyzing individual fractions, and for the first analysis of the pooled fractions, all precursors, including singly charged ions, are allowed to trigger MS2 events. The second acquisition of the pooled fractions is performed considering only multi charged species for fragmentation

9   Protein identification and quantification for N-terminus peptides required the aid of Mascot Distiller software (version 2.5.1) linked to a MASCOT server (version 2.4). The raw data were processed and protein identification and co-/post-translational modifications characterized using the *E. coli* K12 strain reference proteome subset extracted from UniProtKB (version 112), which also included the sequence of the recombinant GNAT proteins. For *Arabidopsis* samples, the data obtained were processed using Mascot Distiller, searching against the Araport-11 protein database, with the parent and fragment mass tolerance, respectively, defined to 10 ppm and 0.5 Da.

However, Mascot software is not able to provide distinct NTA yield for each peptide. The EnCOUNTer tool (version 1.0) (Bienvenut *et al*, 2017b) was used to reprocess the quantified values accessible from the Mascot Distiller xml export files and to supply a final list of the different peptides associated with their NTA yield if relevant data are available to perform the quantification. All further statistical analyses for the N-terminal acetylome were performed using available built-in tools in Microsoft Excel.

## Data availability

Mass spectrometry proteomics data are deposited in the ProteomeXchange Consortium (http://proteomecentral.protemeexchange.org) via the JPOST repository (Deutsch *et al*, 2017) with the dataset identifier PXD015875 (http://www.ebi.ac.uk/pride/archive/projects/PXD015875) and PRIDE repository (https://www.ebi.ac.uk/pride/) with the dataset identifiers PXD016205 (http://www.ebi.ac.uk/pride/archive/projects/PXD016205) and PXD016496 (http://www.ebi.ac.uk/pride/archive/projects/PXD016496).

**Expanded View** for this article is available online.

## Acknowledgements

We warmly thank D. Gibbs (Birmingham University) and T. Arnesen for discussions and critical reading of the manuscript. We thank M. Bilong for final preparation of GNAT samples for GAP analysis, M. Hartl for the initial KAT search in *Arabidopsis*, and P. Pieloch for technical assistance. This project was carried out within the ERA-CAPS Research Programme "KatNat" (IF, CG, PM, MW). The study was funded by the French Agence Nationale de la Recherche agency (ANR-13-BSV6-0004, ANR-17-CAPS-0001-01) to C.G.'s team, the

Deutsche Forschungsgemeinschaft (FI 1655/4-1, INST 211/744-1 FUGG) to I.F., and SFB 1036 TP13 to M.W. and R.H, the "Professorinnenprogramm" of the University of Muenster to I.F. and A.B., the Academy of Finland (330083, 307335, and 321616) to P.M., M.M.K., and A.I., and Doctoral Programme in Molecular Life Sciences at the University of Turku (M.M.K. and A.I.). CG team benefitted from the support of the Labex Saclay Plant Sciences-SPS (ANR-10-LABX-0040-SPS) and used the facilities and expertise of the I2BC proteomic platform SICaPS, supported by IBiSA, Ile de France Region, Plan Cancer, CNRS, and Paris-Saclay University.

## Author contributions

IF and CG headed and supervised the research; WVB, AB, J-BB, CD, JE, IL, JSM, LKS, JS, DS, MMK, AI, EL, TVD, VJ, and GB performed research; WVB, AB, J-BB, IL, CD, IF, MW, RH, PM, TM, CG analyzed the data; CG, TM and IF wrote the paper with assistance of all co-authors.

## Conflict of interest

The authors declare that they have no conflict of interest.

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
