## [Review Process File · Molecular Systems Biology]

Dual lysine and N-terminal acetyltransferases reveal the complexity underpinning protein acetylation

Willy Bienvenut, Annika Brünje, Jean-Baptiste Boyer, Jens Mühlenbeck, Gautier Bernal, Ines Lassowskat, Cyril Dian, Eric Linster, Trinh Dinh, Minna Koskela, Vincent Jung, Julian Seidel, Laura Schyrba, Aiste Ivanauskaitė, Jürgen Eirich, Rüdiger Hell, Dirk Schwarzer, Paula Mulo, Markus Wirtz, Thierry Meinzel, Carmela Giglione, and Iris Finkemeier

DOI: [10.15252/msb.20209464](https://doi.org/10.15252/msb.20209464)

Corresponding author(s): Iris Finkemeier (iris.finkemeier@uni-muenster.de), Carmela Giglione (carmela.giglione@i2bc.paris-saclay.fr)

Review Timeline:

Submission Date:	24th Jan 20
Editorial Decision:	24th Feb 20
Revision Received:	6th May 20
Editorial Decision:	13th May 20
Revision Received:	18th May 20
Accepted:	20th May 20

Editor: Maria Polychronidou

Transaction Report:

24th Feb 2020

Manuscript Number: MSB-20-9464

Title: Dual lysine and N-terminal acetyltransferases reveal the complexity underpinning protein acetylation

Thank you again for submitting your work to Molecular Systems Biology. We have now heard back from the three referees who agreed to evaluate your study. Overall, the reviewers think that the presented findings are novel and relevant for the field. They raise however a series of concerns, which we would ask you to address in a revision.

Without repeating all the points listed below, some of the more fundamental issues raised are the following:

- Reviewer #1 refers to the need to provide further biochemical validations.
- Reviewer #2 points out that some follow up analysis on the biological significance of the dual specificity acetyltransferases would significantly enhance the impact of the study. Reviewer #3 expresses concerns along those lines, as they mention that the study is largely focused on in vitro data. I understand that this is a potentially challenging issue to address and I would be open to discussing in further detail potential suggestions you may have on how to address it.

All other issues raised by the reviewers would need to be convincingly addressed. Please let me know in case you would like to discuss any of the issues raised by the reviewers.

On a more editorial level, we would ask you to address the following issues:

- We have replaced Supplementary Information by the Expanded View (EV format). In this case, all additional figures and tables can be provided as EV Figures. EV Figures should be cited as 'Figure EV1, Figure EV2' etc... in the text and their respective legends should be included in the main text after the legends of regular figures. EV Tables should be provided as individual files, each containing the legend/description of the respective EV Table. For detailed instructions regarding expanded view please refer to our Author Guidelines: . In case several additional EV Figures are generated during revision (total number > 7), we would ask you to provide them in an Appendix PDF. Appendix figures should be labeled and called out as: "Appendix Figure S1, Appendix Figure S2... Appendix Table S1..." etc. Each legend should be below the corresponding Figure/Table in the Appendix. Please include a Table of Contents in the beginning of the Appendix.
- Please provide a "standfirst text" summarizing the study in one or two sentences (approximately 250 characters), three to four "bullet points" highlighting the main findings and a "synopsis image" (550px width and max 400px height, jpeg format) to highlight the paper on our homepage.
- All Materials and Methods need to be described in the main text. We would encourage you to use 'Structured Methods', our new Materials and Methods format. According to this format, the Material and Methods section should include a Reagents and Tools Table (listing key reagents,

experimental models, software and relevant equipment and including their sources and relevant identifiers) followed by a Methods and Protocols section in which we encourage the authors to describe their methods using a step-by-step protocol format with bullet points, to facilitate the adoption of the methodologies across labs. More information on how to adhere to this format as well as downloadable templates (.doc or .xls) for the Reagents and Tools Table can be found in our author guidelines: . An example of a Method paper with Structured Methods can be found here: .

- When you resubmit your manuscript, please download our CHECKLIST (<http://bit.ly/EMBOPressAuthorChecklist>) and include the completed form in your submission. *Please note* that the Author Checklist will be published alongside the paper as part of the transparent process (<https://www.embopress.org/page/journal/17444292/authorguide#transparentprocess>).

If you feel you can satisfactorily deal with these points and those listed by the referees, you may wish to submit a revised version of your manuscript. Please attach a covering letter giving details of the way in which you have handled each of the points raised by the referees. A revised manuscript will be once again subject to review and you probably understand that we can give you no guarantee at this stage that the eventual outcome will be favorable.

Best wishes,

Maria

Maria Polychronidou, PhD
Senior Editor
Molecular Systems Biology

If you do choose to resubmit, please click on the link below to submit the revision online *within 90 days*.

Link Not Available

IMPORTANT: When you send your revision, we will require the following items:

1. the manuscript text in LaTeX, RTF or MS Word format
2. a letter with a detailed description of the changes made in response to the referees. Please specify clearly the exact places in the text (pages and paragraphs) where each change has been made in response to each specific comment given
3. three to four 'bullet points' highlighting the main findings of your study
4. a short 'blurb' text summarizing in two sentences the study (max. 250 characters)
5. a 'thumbnail image' (550px width and max 400px height, Illustrator, PowerPoint or jpeg format), which can be used as 'visual title' for the synopsis section of your paper.
6. Please include an author contributions statement after the Acknowledgements section (see <https://www.embopress.org/page/journal/17444292/authorguide>)
7. Please complete the CHECKLIST available at (<http://bit.ly/EMBOPressAuthorChecklist>). Please note that the Author Checklist will be published alongside the paper as part of the transparent process (<https://www.embopress.org/page/journal/17444292/authorguide#transparentprocess>).
8. Please note that corresponding authors are required to supply an ORCID ID for their name upon

submission of a revised manuscript (EMBO Press signed a joint statement to encourage ORCID adoption). (<https://www.embopress.org/page/journal/17444292/authorguide#editorialprocess>)

Currently, our records indicate that the ORCID for your account is 0000-0002-8972-4026.

Link Not Available

The system will prompt you to fill in your funding and payment information. This will allow Wiley to send you a quote for the article processing charge (APC) in case of acceptance. This quote takes into account any reduction or fee waivers that you may be eligible for. Authors do not need to pay any fees before their manuscript is accepted and transferred to the publisher.

REFeree REPORTS

Reviewer #1:

Summary

In this study, Bienvenut. et al. identify some putative chloroplast acetyltransferases by searching for proteins containing GNAT sequence and secondary structure homology and predicted chloroplast localization, using in silico tools like Prosite, TargetP and others. They identify as many as 10 putative candidates, which indeed share similar secondary structure features and key residues with other GNATs, while most of them contain two predicted Ac-CoA binding motifs, which is unusual. Expression of GFP fusion proteins of all these GNATs in Arabidopsis protoplasts confirms that 8 (7?) of 10 show chloroplast localization. Further characterization of these enzymes is carried out by comparing the lysine acetylome and N-terminal acetylome (GAP assay) of E.coli strains overexpressing MBP-GNATs or empty vector, mainly using mass spectrometry. The results are consistent with these GNATs possessing both KAT and NAT activity. Lastly, global NTA analysis comparison between WT and GNAT2 knockout confirms that GNAT2 has NTA activity in vivo.

General remarks

Previous understanding about acetyltransferases in chloroplast is very limited so the novelty of this study is high. Bienvenut. et al. identify new chloroplast GNATs, most of which have not been previously characterized. Most surprisingly is the finding that of these chloroplast GNATs possess both KTA and NAT activity, a dual activity that does not appear to be conserved in higher eukaryotes. Overall, the cell based mass-spec studies seem to largely support the conclusions of the authors, although there is no biochemical validation, which would be required to demonstrate

direct effects (see below). Assuming that the results can be validated biochemically, and that the other relatively minor issues addressed, we believe that the study would be an exciting contribution to the field.

Major points

1. It is unclear from the study as carried out, if the characterized GNATs carry out the KAT and NAT modifications directly or indirectly. This could be resolved if the authors prepared one or more of the plastid GNAT proteins as a recombinant protein and assayed it against an identified N-terminal or lysine sidechain peptides. This would not only address if the observed activities in cells are direct, but also if the catalytic subunits are necessary and sufficient for the two acetylation activities. The requirement for regulatory subunits could be evaluated by assessing the requirement of added cell extract to the recombinant enzyme for peptide acetyltransferase activity.

Related to the issue above, there is concern that the E.coli GAP assays overexpressing GNATs reveal relaxed substrate specificity, while in vivo GNAT2 KO global NTA analysis displayed preferred small residues at downstream positions. In addition, if so many GNATs show such redundant and non-specific substrate specificity similar to archaeal NATs, why does a single GNAT2 KO show a significant decrease in only some types of substrates? These might be very complex questions to address in vivo but the vitro experiments would go a long way to demonstrating that these GNATs do indeed harbor NTA activity.

Minor points

2. line 55-57, the authors should also mention that some NATs function predominantly posttranslationally, like Naa60 (NatF) and Naa80 (NatH).

3. Line 120, The statement at the end of the introduction that states "has far reaching implications for the study of acetylation in eukaryotic organisms." Should be softened to "may have far reaching...." as this dual specificity may not be conserved in higher eukaryotes.

4. Line 189, change "two bases" to "two catalytic residues"

5. Line 201, the authors state "eight of the ten predicted... show plastid-associated localization", but in between Lines 205 - 208, the data suggests "GNAT6 is probably not within chloroplast and GNAT8 and GNAT9 show cytosolic/nuclear localization." So only seven of them appear to have confirmed plastid localization.

6. in Figure EV2, the expression level of all 8 GNAT proteins appear to vary a lot. The authors should discuss how this might affect the data shown in Figure 3.

7. The authors should provide the identity and N terminal sequence of all substrates identified in the GAP assay in the supplementary information section.

8. GNAT2 AT1G32070.2 in Figure1A is indicated by the authors to be reported in the literature to possess both NAT and KAT activity. In light of this previous report, the novelty of this study would be increased if it focused on another member to demonstrate this dual activity.

9. Based on the GAP assay, the plastid GNATs appear to have a significant overlap in NAT specificity (Figure 3B) and GNAT2 is not the most active GNAT (Figure 3A). In light of this, the authors should discuss this result in the context of the results for the GNAT2 KO. Why do the other NATs not compensate when GNAT2 is knocked out?

10. Line 235, the authors state "GNATs acetylate target proteins in their vicinity". It is still unknown if auto lysine acetylation of MBP-GNATs is inter- or intra-molecular. It is safer to say that highly acetylated MBP peptide might be due to the unusually high abundance of MBP present in the overexpressed cells.

11. Line 298, change "and could quantified half of them" to "and could quantify half of them"

12. The data in Figure 4G requires more explanation. Why is there sequence preferences for positions 7-10. This is quite unusual for NTA's. What does the green annotation of threonine residues meant to symbolize?

13. Figure EV4, The authors should add numbers for each peptide substrate identified for GNAT6,4,7,5,10.

14. Line 338 and 339, the authors state that the GNATs have only partial redundancy on NAT activity, which is contradictory to Figure 3B . To evaluate if GNAT2 has its own plastic "unique" or "relaxed" NTA, the authors could take the top hit substrates and test them as substrates for other GNATs in-vitro.

15. Line 348, The authors should add references for the NAT complex assemblies.

16. line 348, "(8 as to 2019, named NatA/ B/ C/ D/ E/ F/ NAA80)", the authors state 8, but only list 7.

17. Line 351, authors should define NME and the requirement for it.

18. Line 354, change "narrower of proteins" to "narrower set of proteins"

19. The first and second paragraph (line 341-372) of the "discussion" sounds like it could fit in the introduction section.

20. Dinh et al 2015 identified NAA70 as AT2G39000, which the authors named NSI/GNAT2, in line 373. However, in Figure 1A, GNAT2 is listed as AT1G32070.2, while AT2G39000.1 is indicated as GNAT4, which is confusing to readers in terms of naming.

21. Line 402, The sentence that reads "Surprisingly, all identified GNATs display a clear dual KAT and NAT activity, closing the debate of the possibility that the same enzyme can have both activities." is misleading. As far as we are aware there is no debate about whether an enzyme can catalyze both NAT and KAT reaction. The debate centers around whether Naa10 can carry out both activities. In addition, as the authors point out, the structures of the yeast and metazoan NAT substrate binding sites are not appropriately configured to accommodate an internal lysine side chain. The plastid GNAT may indeed have a different type of substrate binding site that may allow it to accommodate both N-terminal and internal amino group substrates.

22. The latter part of the sentence that begins on line 430, which reads "Here we show, that NSI/GNAT2 actually displays an additional NTA activity in vivo next to its KA activity, but GNAT2 inactivation differently affects its KA and NTA targets, suggesting a different acetylation recognition mode." is confusing. It is clear that GNAT2 must have different sequence specificity modes since it acetylates one substrate that has no N-terminal residues and another that does, and it does these in different substrates so it will clearly have different effects. Are the authors trying to say something else?

Reviewer #2:

The manuscript titled "Dual lysine and N-terminal 1 acetyltransferases reveal the complexity underpinning protein acetylation" by Bienvenut et al. show enzymatic evidence in Arabidopsis plastids for a new family of GCN5-related N-acetyltransferases, that possess dual substrate specificity for both N- ϵ and α - lysine. The authors' study is significant for several reasons: (1) it is well-documented that distinct acetyltransferases exist for both these substrates, but there is controversy of whether a single enzyme can have dual specificity, (2) the cellular and biological context of lysine acetylation and N-terminal acetylation have unique features, and (3) the functional consequence of acetylation on protein function is less well understood compared to the number of documented sites of modification. Improved characterization of acetyltransferase enzymes and their specificities should support these efforts.

In the authors study, ten putative GNAT proteins were identified by computational predictions, of which 8 were plastid-localized. Using western blotting and global acetylome profiling with quantitative mass spectrometry, the authors documented that six of the eight GNATs display dual acetyltransferase activities. The KA and NAT profiling experiments of *E. coli* lysates derived from GNAT-expressing strains clearly indicated unique substrate specificities (at least among *E. coli* proteins) and relative GNAT activities, but of course these results should be translated to plastids with some caution. Importantly, the authors did directly support these experiments with targeted knockout of GNAT2 in planta, showing reduced levels of both NTA and KA on plastid proteins.

Overall, the authors two-step in silico and experimental validation approaches efficiently identified and evaluated the top candidate enzymes in Arabidopsis. The in-silico sequence and structural conservation analysis was extremely thorough, highlighting shared and unique features of the putative plastid NATs compared to known cytoplasmic counterparts. Moreover, the experimental design was performed with replicate numbers that were fit-for-purpose and with the appropriate controls. The experimental results were clearly explained, and the figures were impactful. Overall, the study was well conducted and definitively addresses the debate of acetyltransferase dual specificity. The only point lacking, as described in detail below, is the lack of biological context for this dual specificity enzymes, for instance, in chloroplasts. If the authors could provide additional insight here, this study would be significantly elevated and should be considered for publication.

Primary comment

While the authors' demonstration of dual specificity KAT/NATs was convincing and significant, the authors demonstrated in a previous study (Koskela et al, 2018) that GNAT2 KO led to reduced KA levels and defective state transitions of the chloroplasts. While the current study now extends these previous findings by determining that GNAT2 KO also leads to reduced NTA levels in chloroplasts and reveals partial GNAT redundancy, the study lacks a defining experimental result that demonstrates the biological significance of dual specificity. I recognize that without direct evidence of substrate binding characteristics, it is difficult to design experiments that distinguish these activities. Yet, based on the authors structural analysis (Fig 5), perhaps they have considered more subtle genetic ablations of GNAT2 in Arabidopsis?

Minor comment

Pg 16, line 54, a missing word or incorrect sentence structure?

Reviewer #3:

The manuscript 'Dual lysine and N-terminal acetyltransferases reveal the complexity underpinning

protein acetylation' by Bienvenut et al. is a comprehensive investigation into a novel group of enzymes found in plastids. This study represents an important step in defining enzymes responsible for one of the most common protein modifications, namely acetylation. The authors find that among 8 identified plastid acetyltransferases, most of them display a quite convincing dual activity targeting both protein N-termini (alpha amino groups) and lysine side chains (epsilon amino groups). The general opinion is that protein acetyltransferases are either NATs catalyzing N-terminal acetylation or KATs catalyzing lysine acetylation. In many cases this is probably also true, but this study leans support to an more open model where the same enzyme may have both activities.

Candidate proteins were expressed in *E. coli* and the resulting acetylomes were analyzed with respect to both lysine and N-terminal acetylation events. Further sequence analysis defined motif signatures resembling known AcCoA binding motifs etc. One enzyme, GNAT2, was more thoroughly studied by including important *in vivo* experiments truly defining this enzyme as both a NAT and a KAT. The weakness of the study is the overrepresentation of *in vitro* data that may be over-interpreted, but the GNAT2 *in vivo* data are convincing and makes the overall presentation of this group of enzymes valid.

Minor issues:

1) The language is mostly good, but some typos and artistic twists here and there might be adjusted for a clearer presentation.

For example,

Line 32: 'supported by' should be rephrased.

Line 298: 'quantified' -> 'quantify'

Line 354: Rephrase 'a narrower of proteins'

2) Line 195: '...lle in GNAT1, a T in GNAT2 and a S in GNAT3...'

Please, consistently use 1-letter or 3-letter aa code throughout the manuscript.

3) Lines 81-88: The reader is left with the impression that (almost) all eukaryotic NATs are active as complexes. Please add another sentence or two mentioning that also many eukaryotic NATs appear to be non-complexed, such as NAA40 which is highly specific for Histones H2A and H4 (Song et al., *J Biol Chem*, 2003, PMID: 12915400), NAA60 which acts on transmembrane proteins (Aksnes et al., *Cell Rep*, 2015, PMID: 25732826) and NAA80 acetylating actins (Drazic et al., *PNAS USA*, 2018, PMID: 29581253).

4) Line 97: Evjenth et al, 2009 refers to the dual capacity of NAA50 to perform both NTA and KA, but the sentence focuses on NAA10, thus either remove this reference or change the sentence.

5) Figure S1. GNAT4 and GNAT7 is duplicated while GNAT3 and GNAT10 are missing.

6) Line 129-130: Please include include info on new/old names correlating the GNAT1-10 nomenclature with NAA70 and NSI enzymes. In Table S1 there is no mention of NAA70 or NSI while the text does not mention GNAT names when listing NAA70 and NSI.

7) According to Figure 1B, only GNAT9 is missing a chloroplastic transit peptide while according to Table S1 GNAT9, GNAT6 and GNAT3 all miss a chloroplastic transit peptide (while having a mitochondrial targeting signal). Table S1 text says: 'Subcellular 52 localization was predicted with Target P. However, only eight of them possess a clear TP.'. Please clarify and unify to avoid confusion.

8) Figure 2A: Please combine this figure with the Coomassie stained gels in Fig EV2 in order to allow the reader to compare the bands. Also, GNAT4 anti-Ac-Lys should be repeated with less loading/less signal to allow for better interpretation.

9) Line 348: There are only 7 NATs in the cytosol of humans (not 8).

10) Line 349: remove 'narrow' since NatA for instance displays a broad substrate specificity.

11) For Discussion: Please revisit the interpretation of findings in Koskela, Plant Cell, 2018 in light of these new findings on the NAT-activity of GNAT2. (activities and mechanisms leading to phenotype etc)

Point-by-point responses to reviewers' comments

Reviewer #1

Summary

In this study, Bienvenut. et al. identify some putative chloroplast acetyltransferases by searching for proteins containing GNAT sequence and secondary structure homology and predicted chloroplast localization, using in silico tools like Prosite, TargetP and others. They identify as many as 10 putative candidates, which indeed share similar secondary structure features and key residues with other GNATs, while most of them contain two predicted Ac-CoA binding motifs, which is unusual. Expression of GFP fusion proteins of all these GNATs in Arabidopsis protoplasts confirms that 8 (7?) of 10 show chloroplast localization. Further characterization of these enzymes is carried out by comparing the lysine acetylome and N-terminal acetylome (GAP assay) of E.coli strains overexpressing MBP-GNATs or empty vector, mainly using mass spectrometry. The results are consistent with these GNATs possessing both KAT and NAT activity. Lastly, global NTA analysis comparison between WT and GNAT2 knockout confirms that GNAT2 has NTA activity in vivo.

General remarks

Previous understanding about acetyltransferases in chloroplast is very limited so the novelty of this study is high. Bienvenut. et al. identify new chloroplast GNATs, most of which have not been previously characterized. Most surprisingly is the finding that of these chloroplast GNATs possess both KTA and NAT activity, a dual activity that does not appear to be conserved in higher eukaryotes. Overall, the cell based mass-spec studies seem to largely support the conclusions of the authors, although there is no biochemical validation, which would be required to demonstrate direct effects (see below). Assuming that the results can be validated biochemically, and that the other relatively minor issues addressed, we believe that the study would be an exciting contribution to the field.

Major points

1. It is unclear from the study as carried out, if the characterized GNATs carry out the KAT and NAT modifications directly or indirectly. This could be resolved if the authors prepared one or more of the plastid GNAT proteins as a recombinant protein and assayed it against an identified N-terminal or lysine sidechain peptides. This would not only address if the observed activities in cells are direct, but also if the catalytic subunits are necessary and sufficient for the two acetylation activities. The requirement for regulatory subunits could be evaluated by assessing the requirement of added cell extract to the recombinant enzyme for peptide acetyltransferase activity.

We would like to thank the reviewer for their comments and suggestions. We selected two GNATs, representing two subtypes from the different branches of the phylogenetic tree (Fig. EV1), for protein purification and *in vitro* activity tests. For the activity assay, we used a HPLC-based method, which allows the detection of the appearance of the acetylated peptide substrate over time. We used standard peptides, which were previously developed for lysine deacetylase as well as acetyltransferase enzyme assays by our collaborators (laboratory of Prof. Dirk Schwarzer, University of Tübingen), who are now included as co-authors on the manuscript. For this study, they additionally synthesized the same peptide with different alpha amino acids possessing either a free or acetylated N-terminus. Hence, these peptides allowed an unambiguous **determination of both N-alpha and ε-lysine activities** on the same peptide sequence. Interestingly, both enzymes were **active without any auxiliary subunits**. Furthermore, the assays revealed that both enzymes have relaxed substrate specificities, with some preferences for particular N-terminal amino acids. This nicely correlates the data obtained from the GAP assay and the study of the N-terminomics analysis of the GNAT knockout line. These new data are now displayed in a new Table (**Table 1**).

Related to the issue above, there is concern that the E.coli GAP assays overexpressing GNATs reveal relaxed substrate specificity, while in vivo GNAT2 KO global NTA analysis displayed preferred small residues at downstream positions. In addition, if so many GNATs show such redundant and non-

specific substrate specificity similar to archaeal NATs, why does a single GNAT2 KO show a significant decrease in only some types of substrates? These might be very complex questions to address in vivo but the in vitro experiments would go a long way to demonstrating that these GNATs do indeed harbor NTA activity.

Concerning the use and the interpretation of the data from the GAP assay, we would like to point out, as we reported in the result section, that this test has been previously validated with cytosolic AtNAA10 (Dinh et al 2015, Proteomics, Figure 3 panels. A&C). Expression of AtNAA10 alone in bacteria fully recapitulates NatA selectivity *in planta* as described in a separate work (Linster et al 2015 Nat. Commun.). This specificity also tightly fits that observed in other organisms. Additionally, we recently characterized AtNAA60 and AtNAA50; both studies are now under minor revision in two different journals and the two manuscripts include a GAP assay for both enzymes. The data reveal similar substrate specificities with known orthologous NAAs. We assume that the GAP assay as a result is very robust, relevant and reflecting the biological observations. Its unique advantage is to sample a large array of sequences at once without any of the a priori and limitations which biochemical studies usually direct (choice of peptide sequence, limited sequence corpus, additional chemical groups for detection...).

Concerning the issue of relaxed substrate specificity of plastid GNATs observed with the GAP assay, there is kind of a misunderstanding and our presentation was likely misleading. We now better address this issue in the new result section. In brief, the GAP assay revealed that the NTA substrate specificity of any of the plastid GNATs gathered NatA/B/C/D/E specificities. Such specificities were classified as mainly depending on the nature of aminoacids at positions 1>2>3. For instance, all of six most active plastid *At*GNATs were very efficient for N-termini starting with an initiator Met, but with clear differences induced by the amino acid at position 2. To better illustrate this issue, we decided to increase the threshold criteria from 2% to 5% to display them in the new **Figure 3A** for each GNAT. In addition, we now include a new dashed line showing how many substrates are N-acetylated with a value above 30%, which is very significant. The complete dataset is now available in **Dataset EV3**. Both the new **Figure 4A** and **Dataset EV3** now nicely illustrate that some sequences are only recognized by one GNAT and not by the others, while some others are recognized by several GNATs. A few are even N- α -acetylated by all of them. Though our GAP data cover almost 400 sequences, we have too little information to conclude fully on the specificity of each GNAT. An attempt towards this end is given in Figure 4B where we built a logo illustrating *At*GNAT2 specificity with respect to the seven others *At*GNATs; this reveals unique features which are now discussed.

Altogether, the data reveal that plastid GNATs have only partial overlapping relaxed substrate specificity. Because other plastid GNATs are most likely able to still acetylate plastid proteins in the context of the *At*GNAT2 mutant (*nsi-1*), the characterization of this mutant reflects mostly the most sensitive *At*GNAT2 NTA- and KA-substrates rather than its substrate specificity. Furthermore, the NTA of different N-termini of the same proteins, such as F16P1, were affected in *At*GNAT2-defective plant lines, suggesting long distance contacts between *At*GNAT2 and their substrates. This can explain the selectivity of *At*GNAT2 for specific substrates. The new **Dataset EV3** shows all GAP data while Figure 4 illustrates some examples addressing the issue related to “overlapping substrate specificity”. We also discuss and compare **Figure 4B** and Figure 5G and reveal similar features.

Minor points

2. line 55-57, the authors should also mention that some NATs functional predominantly posttranslationally, like Naa60 (NatF) and Naa80 (NatH).

Post-translational NTA by NatF and NatH are now quoted (line 76-78) and reference Aksnes et al. Mol Cell 2019 was added to complete the references reported.

3. Line 120, The statement at the end of the introduction that states "has far reaching implications for the study of acetylation in eukaryotic organisms." Should be softened to "may have far reaching...." as this dual specificity may not be conserved in higher eukaryotes.

This sentence has been softened as suggested.

4. Line 189, change "two bases" to "two catalytic residues"

This has been changed accordingly.

5. Line 201, the authors state "eight of the ten predicted... show plastid-associated localization", but in between Lines 205 - 208, the data suggests "GNAT6 is probably not within chloroplast and GNAT8 and GNAT9 show cytosolic/nuclear localization." So only seven of them appear to have confirmed plastid localization.

This is correct, only seven of them are localized within plastids. However, the term plastid-associated localization is not dedicated only for proteins that are localized inside plastids but applies to all that are connected to the organelle; the same is valid for other compartments. GNAT6 was predicted as an organellar-localized protein and a transit peptide was also predicted for this protein. In our GFP-images, GNAT6 was observed as associated to chloroplast. Thus, eight plastid-associated GNATs is the correct term to use. Additionally, we have repeated the localization analysis for GNAT6 with some subcellular localization markers. Importantly, the GNAT6-GFP signal does not overlap with mitochondria. In addition to the chloroplast-associated localization we now confirmed some additional nuclear localization for this protein, which is shown by co-expression with a nuclear envelope marker protein.

6. in Figure EV2, the expression level of all 8 GNAT proteins appear to vary a lot. The authors should discuss how this might affect the data shown in Figure 3.

The expression level of GNATs does not seem to influence the NTA and KA activity. For instance, a low level of both NTA and KA activity was observed with GNAT1 even if this protein is the most highly expressed in bacteria. In contrast, GNAT4 was among the less expressed GNATs but its NTA and KA activity was among the highest one. We have now indicated in **Figure 3A** the relative level of expression of each GNAT just under the panel that displays the number of substrates. GNAT1 is the best expressed but instead it modifies the lowest number of proteins with high efficiency (see dashed line); this is in contrast to GNAT5. This is discussed now lines 335-338.

7. The authors should provide the identity and N terminal sequence of all substrates identified in the GAP assay in the supplementary information section.

A new dataset of the identified *E.coli* N-termini in the GAP assay is now provided as **Dataset EV3**. This is also mentioned in the manuscript line 321.

8. GNAT2 AT1G32070.2 in Figure 1A is indicated by the authors to be reported in the literature to possess both NAT and KAT activity. In light of this previous report, the novelty of this study would be increased if it focused on another member to demonstrate this dual activity.

Except GNAT4, none of the identified plastid GNATs was known to have both NTA and auto-KA activities; GNAT2 was previously shown by us to display significant KA activity (Koskela 2018 TPC) in addition to low serotonin acetyltransferase activity (Lee et al. 2014, PMID: 25250906) as now cited in the legend to the figure (line 1307). The NTA activity of GNAT2 has never been reported either *in vitro* or *in vivo*.

9. Based on the GAP assay, the plastid GNATs appear to have a significant overlap in NAT specificity (Figure 3B) and GNAT2 is not the most active GNAT (Figure 3A). In light of this, the authors should discuss this result in the context of the results for the GNAT2 KO. Why do the other NATs not compensate when GNAT2 is knocked out?

Please see answer to major point 1.

10. Line 235, the authors state "GNATs acetylate target proteins in their vicinity". It is still unknown if auto lysine acetylation of MBP-GNATs is inter- or intra-molecular. It is safer to say that highly acetylated MBP peptide might be due to the unusually high abundance of MBP present in the overexpressed cells.

The sentence has been changed accordingly.

11. Line 298, change "and could quantified half of them" to "and could quantify half of them"

The sentence has been changed accordingly.

12. The data in Figure 4G requires more explanation. Why is there sequence preferences for positions 7-10. This is quite unusual for NTA's.

As discussed before, GNAT2 displays unambiguous specificity but it is still unclear where it comes from and it is unrelated to positions 1-3. We also know that there is specificity beyond the first residues as the same protein with two different N-termini is a substrate of GNAT2. We thus have considered looking at the first 10 residues to reveal whether there is something appearing downstream. The sequence logo in Figure 4G (now **Figure 5G**) shows no strong specificity appearing clearly as only Thr or Leu are retrieved in less than 1/3 of the 10 affected proteins. A comparison with the data discussed with the new **Figure 4B** is now available, showing similar features relative to GNAT2 specificity.

What does the green annotation of threonine residues meant to symbolize?

The color symbol is associated to the default choice, which the software Icelogo proposes for each class of amino acid. Green is for the class of small hydrophilic uncharged residues including Ser, Thr, Gly... Asp or Glu would be red. Lys, Arg, His would be blue. Asn or Gln, purple. Hydrophobic (Ala, Ile, Val, Leu, Trp, Tyr, Phe, Pro) residues are black. This is now indicated in the legend to the new **Figure 4B** where all colors are retrieved and reminded in the legend of this figure (now **Figure 5**).

13. Figure EV4, The authors should add numbers for each peptide substrate identified for GNAT6,4,7,5,10.

Numbers of each peptide substrate identified for the different GNAT are now reported in the new version (now **Figure EV6**).

14. Line 338 and 339, the authors state that the GNATs have only partial redundancy on NAT activity, which is contradictory to Figure 3B. To evaluate if GNAT2 has its own plastic "unique" or "relaxed" NTA, the authors could take the top hit substrates and test them as substrates for other GNATs in-vitro.

A new **Figure 4** and **Dataset EV3** have been prepared according to the reviewer's suggestion, which better highlight partial overlapping substrate specificity. See also answer to major points.

15. Line 348, The authors should add references for the NAT complex assemblies.

We have added Aksnes et al (2019) Mol Cell.

16. line 348, "(8 as to 2019, named NatA/ B/ C/ D/ E/ F/ NAA80)", the authors state 8, but only list 7.

Number has been corrected.

17. Line 351, authors should define NME and the requirement for it.

NME and its requirement have been added in addition to two references.

18. Line 354, change "narrower of proteins" to "narrower set of proteins"

Done

19. The first and second paragraph (line 341-372) of the "discussion" sounds like it could fit in the introduction section.

We preferred to have a more balanced introduction pointing on the GNAT family to which KAT and NAT enzymes belong.

20. Dinh et al 2015 identified NAA70 as AT2G39000, which the authors named NSI/GNAT2, in line 373. However, in Figure 1A, GNAT2 is listed as AT1G32070.2, while AT2G39000.1 is indicated as GNAT4, which is confusing to readers in terms of naming.

AT2G39000 corresponds to NAA70 and now GNAT4, whereas we clearly stated that AT1G32070.2 was previously named NSI and now GNAT2. Figure and text are correct.

21. Line 402, The sentence that reads "Surprisingly, all identified GNATs display a clear dual KAT and NAT activity, closing the debate of the possibility that the same enzyme can have both activities." is misleading. As far as we are aware there is no debate about whether an enzyme can catalyze both NAT and KAT reaction. The debate centers around whether Naa10 can carry out both activities. In addition, as the authors point out, the structures of the yeast and metazoan NAT substrate binding sites are not appropriately configured to accommodate an internal lysine side chain. The plastid GNAT may indeed have a different type of substrate binding site that may allow it to accommodate both N-terminal and internal amino group substrates.

The last part of the sentence has been omitted to avoid this misleading presentation (see lines 114-131).

22. The latter part of the sentence that begins on line 430, which reads "Here we show, that NSI/GNAT2 actually displays an additional NTA activity in vivo next to its KA activity, but GNAT2 inactivation differently affects its KA and NTA targets, suggesting a different acetylation recognition mode." is confusing. It is clear that GNAT2 must have different sequence specificity modes since it acetylates one substrate that has no N-terminal residues and another that does, and it does these in different substrates so it will clearly have different effects. Are the authors trying to say something else?

Since the most affected KA and NTA proteins in the GNAT2 mutant context are different, it is most likely that the recognition mode of GNAT2 for them is different. We changed the sentence to avoid confusion with the activity (lines 522-524).

Reviewer #2:

The manuscript titled "Dual lysine and N-terminal 1 acetyltransferases reveal the complexity underpinning protein acetylation" by Bienvenut et al. show enzymatic evidence in Arabidopsis plastids for a new family of GCN5-related N-acetyltransferases, that possess dual substrate specificity for both N- α and ϵ -lysine. The authors' study is significant for several reasons: (1) it is well-documented that distinct acetyltransferases exist for both these substrates, but there is controversy of whether a single enzyme can have dual specificity, (2) the cellular and biological context of lysine acetylation and N-terminal acetylation have unique features, and (3) the functional consequence of acetylation on protein function is less well understood compared to the number of documented sites of modification. Improved characterization of acetyltransferase enzymes and their specificities should support these efforts.

In the authors study, ten putative GNAT proteins were identified by computational predictions, of which 8 were plastid-localized. Using western blotting and global acetylome profiling with quantitative mass spectrometry, the authors documented that six of the eight GNATs display dual acetyltransferase activities. The KA and NAT profiling experiments of E. coli lysates derived from GNAT-expressing strains clearly indicated unique substrate specificities (at least among E. coli proteins) and relative GNAT activities, but of course these results should be translated to plastids with some caution. Importantly, the authors did directly support these experiments with targeted knockout of GNAT2 in planta, showing reduced levels of both NTA and KA on plastid proteins.

Overall, the authors two-step in silico and experimental validation approaches efficiently identified and evaluated the top candidate enzymes in Arabidopsis. The in-silico sequence and structural conservation analysis was extremely thorough, highlighting shared and unique features of the putative plastid NATs compared to known cytoplasmic counterparts. Moreover, the experimental design was performed with replicate numbers that were fit-for-purpose and with the appropriate controls. The experimental results were clearly explained, and the figures were impactful. Overall, the study was well conducted and definitively addresses the debate of acetyltransferase dual specificity. The only point lacking, as described in detail below, is the lack of biological context for this dual specificity enzymes, for instance, in chloroplasts. If the authors could provide additional insight here, this study would be significantly elevated and should be considered for publication.

Primary comment

While the authors' demonstration of dual specificity KAT/NATs was convincing and significant, the authors demonstrated in a previous study (Koskela et al, 2018) that GNAT2 KO led to reduced KA levels and defective state transitions of the chloroplasts. While the current study now extends these previous findings by determining that GNAT2 KO also leads to reduced NTA levels in chloroplasts and reveals partial GNAT redundancy, the study lacks a defining experimental result that demonstrates the biological significance of dual specificity. I recognize that without direct evidence of substrate binding characteristics, it is difficult to design experiments that distinguish these activities. Yet, based on the authors structural analysis (Fig 5), perhaps they have considered more subtle genetic ablations of GNAT2 in Arabidopsis?

We do appreciate the reviewer's comment, however, to answer this most interesting and intriguing issue this will require a dedicated study of probably several years due to the time it takes for plant complementation and mutation analyses. This goes well beyond this study. For instance, the *nsi* knockout induces a photosynthetic phenotype. In Koskela et al (2018), we showed that the KA status of protein components belonging to PSI, PSII and LHC were modified in the KO compared to the WT. In the Koskela (2018) paper, we additionally demonstrated that deletion of the enzyme GNAT2 (NSI) resulted in a complete loss of the ability of Arabidopsis plants to perform state transitions and more recently that the loss of NSI/GNAT2 had no effect on plant growth under control growth conditions, but it was severely affected when plants were subjected to fluctuating light conditions (Koskela et al, 2020). Deletion of the enzyme GNAT2 (NSI) was shown to induce a decrease on KA, but not phosphorylation of the LHCII proteins (Koskela et al, 2018). Among the affected proteins in the NTA acetylome of the present study, we did not retrieve any protein from the light reactions such as LHCII, which was found as a target protein for Lys-acetylation of NSI. Hence, the identified NTA substrates of NSI do not support the role for NTA acetylation on the photosynthesis phenotype. Altogether, these studies suggest that KA is a major modification influencing state transition plastid. Of course, these studies are not exhaustive and a further deeper characterization of isolated PSII-LCHII complexes under different light conditions will be needed to fully clarify the impact of KA and NTA activity of GNAT2 and even other plastid GNATs in photosynthesis. Concerning more subtle genetic ablations, we have not in hands yet the 3D structure of GNAT2 or any other plastid GNAT, which we would need for a targeted analysis. Indeed, plastid GNATs belong to the NAA family but it remains impossible – even using 3D models derived from known structures - to predict the various substitutions which would make “subtle” alterations such as blocking Lys with respect to N-terminal acetylation or vice versa. Our data indicate that plastid GNATs are part of a new NAA family which

covers at least two subtypes, GNAT1/2/3 and GNAT4/5/6/7 and 10 (now displayed in **Figure EV1**). Hence, deeper investigations will keep us and others busy for the coming years.

Minor comment

Pg 16, line 54, a missing word or incorrect sentence structure?

Unfortunately, on page 16 there is no line 54, hence we do not know which sentence you are referring to. The manuscript was checked again for any grammar or spelling mistakes. We hope we detected all of them and removed them now.

Reviewer #3:

The manuscript 'Dual lysine and N-terminal acetyltransferases reveal the complexity underpinning protein acetylation' by Bienvenut et al. is a comprehensive investigation into a novel group of enzymes found in plastids. This study represents an important step in defining enzymes responsible for one of the most common protein modifications, namely acetylation. The authors find that among 8 identified plastid acetyltransferases, most of them display a quite convincing dual activity targeting both protein N-termini (alpha amino groups) and lysine side chains (epsilon amino groups). The general opinion is that protein acetyltransferases are either NATs catalyzing N-terminal acetylation or KATs catalyzing lysine acetylation. In many cases this is probably also true, but this study leans support to an more open model where the same enzyme may have both activities.

Candidate proteins were expressed in E. coli and the resulting acetylomes were analyzed with respect to both lysine and N-terminal acetylation events. Further sequence analysis defined motif signatures resembling known AcCoA binding motifs etc. One enzyme, GNAT2, was more thoroughly studied by including important in vivo experiments truly defining this enzyme as both a NAT and a KAT. The weakness of the study is the overrepresentation of in vitro data that may be over-interpreted, but the GNAT2 in vivo data are convincing and makes the overall presentation of this group of enzymes valid.

Minor issues:

1) The language is mostly good, but some typos and artistic twists here and there might be adjusted for a clearer presentation.

For example,

Line 32: 'supported by' should be rephrased.

Line 298: 'quantified' -> 'quantify'

Line 354: Rephrase 'a narrower of proteins'

Typos and unclear sentences have been removed or clarified, accordingly.

2) Line 195: '...Ile in GNAT1, a T in GNAT2 and a S in GNAT3...'
Please, consistently use 1-letter or 3-letter aa code throughout the manuscript.

We have corrected this issue.

3) Lines 81-88: The reader is left with the impression that (almost) all eukaryotic NATs are active as complexes. Please add another sentence or two mentioning that also many eukaryotic NATs appear to be non-complexed, such as NAA40 which is highly specific for Histones H2A and H4 (Song et al., J Biol Chem, 2003, PMID: 12915400), NAA60 which acts on transmembrane proteins (Aksnes et al., Cell Rep, 2015, PMID: 25732826) and NAA80 acetylating actins (Drazic et al., PNAS USA, 2018, PMID: 29581253).

Sentences and references have been added to avoid the misleading presentation as suggested.

4) Line 97: *Evjenth et al, 2009 refers to the dual capacity of NAA50 to perform both NTA and KA, but the sentence focuses on NAA10, thus either remove this reference or change the sentence.*

Two sentences are now added to indicate first that NAA40, 50 and 60 have been suggested to have low KA activity, in addition to their high NTA activity (lines 115-117). The second sentence focuses on NAA10.

5) *Figure S1. GNAT4 and GNAT7 is duplicated while GNAT3 and GNAT10 are missing.*

We thank the reviewer for pointing out this issue. We do apologize and we have now corrected all labels, dealing with GNATs. This confusion is mainly due to the new nomenclature, which we have adopted for the purpose of the publication and we mixed up new and old for some reason. We also corrected the legend to fit with the proper Arabidopsis entries.

6) *Line 129-130: Please include info on new/old names correlating the GNAT1-10 nomenclature with NAA70 and NSI enzymes. In Table S1 there is no mention of NAA70 or NSI while the text does not mention GNAT names when listing NAA70 and NSI.*

Old and new nomenclatures are now reported in the legend to **Figure EV1**.

7) *According to Figure 1B, only GNAT9 is missing a chloroplastic transit peptide while according to Table S1 GNAT9, GNAT6 and GNAT3 all miss a chloroplastic transit peptide (while having a mitochondrial targeting signal). Table S1 text says: 'Subcellular 52 localization was predicted with Target P. However, only eight of them possess a clear TP..'. Please clarify and unify to avoid confusion.*

As explained in M&M, when we searched for plastid NAT/KAT, we used parallel *in silico* strategies, including subcellular prediction tools. The prediction tools are very powerful but far to be robust particularly with plants. For instance, Target P encounters frequent erroneous predictions between mitochondrial (Mt) and chloroplast (Cp) localisations. Thus, we considered both Cp and Mt predicted candidates. In the independent search of KAT, we retrieved a list of 35 Arabidopsis candidates but only 10 (the same retrieved as in the NAT search), which contained a putative organellar targeting peptide according to Target P-1.1 predictor. In Table EV1 (even if a Mt localization was predicted for GNAT3, GNAT6 and GNAT9) only GNAT 9 missed a TP accordingly to a more in-depth analysis using N-TerPred, ChloroP, and multiple sequence alignment of plant orthologues. This is more clearly indicated in the table now.

8) *Figure 2A: Please combine this figure with the Coomassie stained gels in Fig EV2 in order to allow the reader to compare the bands. Also, GNAT4 anti-Ac-Lys should be repeated with less loading/less signal to allow for better interpretation.*

We now have combined both figures and included a less exposed blot for GNAT4, in addition to the overexposed blot. In comparison to the other GNATs, the luminescence signal of the acetylated proteins was already overexposed after a few seconds, indicating the strong activity of this enzyme, which was confirmed in the quantitative mass spectrometry analysis.

9) *Line 348: There are only 7 NATs in the cytosol of humans (not 8).*

This has been corrected.

10) *Line 349: remove 'narrow' since NatA for instance displays a broad substrate specificity.*

“Narrow” has been changed with “well-defined”.

11) *For Discussion: Please revisit the interpretation of findings in Koskela, Plant Cell, 2018 in light of*

these new findings on the NAT-activity of GNAT2. (activities and mechanisms leading to phenotype etc)

We now have extended the discussion regarding the photosynthetic phenotype of GNAT2. However, in contrast to the detected substrates for K-acetylation, none of the detected NTA substrates of GNAT2 allows a direct explanation for the observed state transition phenotype.

13th May 2020

Manuscript Number: MSB-20-9464R

Title: Dual lysine and N-terminal acetyltransferases reveal the complexity underpinning protein acetylation

Thank you for sending us your revised manuscript. We have now heard back from reviewer #1 who was asked to evaluate your revised study. As you will see below, the reviewer is satisfied with the modifications made and thinks that the study is now suitable for publication. As such, I am glad to inform you that we can soon accept your manuscript for publication, pending some minor editorial issues listed below.

- Our data editors have noticed some unclear or missing information in the figure legends, please see the attached .doc file. Please make all requested text changes using the attached file and *keeping the "track changes" mode* so that we can easily access the edits made.

- There is a callout to Dataset EV6, but no Dataset EV6 exists. Could you please edit the callout accordingly?

- The EV Dataset and EV Table legends can be removed from the main text. Please provide them only in the respective .xls files.

- Please provide high-resolution individual files for each of the main and EV figures.

Please resubmit your revised manuscript online ****within one month**** and ideally as soon as possible. If we do not receive the revised manuscript within this time period, the file might be closed and any subsequent resubmission would be treated as a new manuscript. Please use the Manuscript Number (above) in all correspondence.

Click on the link below to submit your revised paper.

Link Not Available

Kind regards,

Maria

Maria Polychronidou, PhD
Senior Editor
Molecular Systems Biology

If you do choose to resubmit, please click on the link below to submit the revision online before 12th Jun 2020.

Link Not Available

Please note that corresponding authors are required to supply an ORCID ID for their name upon submission of a revised manuscript (EMBO Press signed a joint statement to encourage ORCID adoption) (<https://www.embopress.org/page/journal/17444292/authorguide#editorialprocess>).

Currently, our records indicate that the ORCID for your account is 0000-0002-8972-4026.

Link Not Available

The system will prompt you to fill in your funding and payment information. This will allow Wiley to send you a quote for the article processing charge (APC) in case of acceptance. This quote takes into account any reduction or fee waivers that you may be eligible for. Authors do not need to pay any fees before their manuscript is accepted and transferred to the publisher.

REFEREE REPORTS

Reviewer #1:

The authors have satisfactorily addressed our major and minor concerns in the revised manuscript. They have performed in vitro assays to confirm that GNAT2 and GNAT10 each independently can have acetylation activity toward both N termini and internal lysine substrates. The authors also made new figures to demonstrate more clearly the substrate overlap of these GNATs. In addition, minor issues have been addressed. We feel that the study is now suitable for publication.

The Authors have made the requested editorial changes.

20th May 2020

Manuscript number: MSB-20-9464RR

Title: Dual lysine and N-terminal acetyltransferases reveal the complexity underpinning protein acetylation

Thank you for sending us your revised manuscript and for performing the final requested minor changes. We are now satisfied with the modifications made and I am pleased to inform you that your paper has been accepted for publication.

Corresponding Author Name: Dr. Carmela Giglione, Dr. Iris Finkemeier

Journal Submitted to: Mol sys Biol

Manuscript Number: MSB-20-9464